# Time series Foundation Models based on Physics-Informed Synthetic Histories for Cold-Start Photovoltaic Forecasting

**Lorenzo Longarini** [* 1]   **Alessandro Rongoni** [* 1]   **Simone Silenzi** [* 1 2]   **Emanuele Frontoni** [3]   **Riccardo Rosati** [3]

## Abstract

At commissioning time, Photovoltaic (PV) operators must forecast production before target-site observations are available, limiting the direct use of standard supervised forecasters. This cold-start setting is addressed with a zero-shot pipeline that generates a synthetic production history from plant metadata and meteorological covariates, enabling time-series foundation models (TSFMs) to forecast through inference-time conditioning. Five TSFMs are benchmarked against classical baselines under strict Cold-Start Baseline, Real Feedback, and Self-Forecast Feedback strategies. The evaluation spans 440 PV sites across four datasets and diverse climates regimes. Covariate-aware foundation models outperform baselines by approximately $1.7$–$2\times$: TabPFN-TS achieves the lowest error under Real Feedback (MAE $0.514$, RMSE $0.721 \, \mathrm{kW\,h\,kWp^{-1}\,d^{-1}}$), while Chronos-2 is most robust under Self-Forecast Feedback. Performance is largely insensitive to the synthetic-history source, indicating that accuracy is driven more by the availability of plausible temporal context than by the specific generator.

## 1. Introduction

Photovoltaic (PV) operators require day-0 production forecasts at commissioning time for grid integration, self-consumption, battery sizing, and asset management. As no historical observations from the target system are available, models must rely exclusively on plant metadata and exogenous meteorological covariates, precluding the direct use of supervised approaches conditioned on past target observations. Cold-start PV forecasting has been primarily addressed through transfer learning from data-rich or neighbouring sites, including LSTM initialisation, feature extraction, and selective parameter freezing (Sarmas et al., 2022), cross-learning across panels indexed by shared metadata (Bottieau et al., 2022), and cross-continental Transformer transfer (Bak et al., 2025). Hybrid physical and machine learning approaches have also been proposed, coupling calibrated physical model chains with domain-adversarial training (Liao et al., 2024). A closely related paradigm is introduced by SolNet (Depoortere et al., 2024). In this approach, a PVGIS-derived synthetic history is used to fine-tune a task-specific LSTM on the target plant. PVGIS is a European Commission photovoltaic performance and irradiation database that provides model-based production estimates from geospatial and meteorological inputs. However, this approach remains tied to a single surrogate source and a specific supervised architecture. Complementary approaches include hybrid physics and machine learning models (Santos et al., 2024; Pombo et al., 2022) and purely data-driven synthetic generators based on GANs (Chen et al., 2018) or diffusion models (Lin et al., 2024), which require pre-existing real datasets for training. Within this space, optimised physical model chains implemented in open-source libraries such as pvlib (Holmgren et al., 2018) remain comparatively under-explored as synthetic data sources for cold-start forecasting. These approaches highlight a common limitation: effective performance typically relies on either access to real target data, explicit supervised adaptation, or tightly coupled modelling assumptions. Recent advances in time-series foundation models (TSFMs) provide an alternative paradigm, enabling forecasting through zero-shot inference conditioned on contextual sequences rather than parameter updates. Their effectiveness has been demonstrated across domains, as confirmed by unified benchmarks (Li et al., 2024) and energy applications such as load forecasting (Meyer et al., 2024; Simeone, 2026). This formulation is particularly suited to cold-start conditions, where no target observations are available. Existing FM-based solar forecasting studies typically focus on short-term horizons and rely on auxiliary sensing streams, such as satellite imagery (Dong et al., 2025), sky-camera feeds (Mishra et al.,

---

[*]Equal contribution   [1]Sistemi 2000 srl, Civitanova Marche, Italy   [2]Department of Engineering "Enzo Ferrari", University of Modena and Reggio Emilia, Modena, Italy   [3]Department of Political Sciences, Communication and International Relations, University of Macerata, Macerata, Italy. Correspondence to: Lorenzo Longarini <lorenzo.longarini@gmail.com>, Alessandro Rongoni <alessandro.rongoni2000@gmail.com>, Simone Silenzi <s.silenzi1@gmail.com>.

*Proceedings of the $2^{nd}$ ICML Workshop on Foundation Models for Structured Data*, Seoul, South Korea. 2026. Copyright 2026 by the author(s).

2025), or language-model-based reprogramming pipelines (Liu et al., 2026). However, it remains unclear whether synthetic production histories can provide a reliable temporal context for TSFMs under realistic cold-start PV conditions. To address this gap, cold-start PV forecasting is formulated as a two-stage zero-shot pipeline: a synthetic-history generator constructs a surrogate production sequence from commissioning-time metadata and meteorology, and a downstream forecaster uses this sequence as temporal context without site-specific parameter updates. Within this framework, three contributions are provided: (i) a systematic zero-shot evaluation of heterogeneous TSFM architectures across 440 sites spanning multiple climate regimes; (ii) a set of *context strategies* that isolate strict cold-start performance from the effects of real telemetry and autoregressive self-conditioning; and (iii) an open, physics-based synthetic generator that enables reproducible and source-independent construction of temporal context. This design enables controlled analysis of model behaviour across distinct operational regimes.

## 2. Material and Methods

Cold-start PV forecasting requires predictions before plant-specific production histories are available. We address this setting with a two-stage pipeline (Fig. 1): a synthetic-history source generates a surrogate production sequence from metadata and meteorological covariates, and a downstream forecaster uses it as temporal context.

### 2.1. Problem formulation

At commissioning time, only plant metadata $\mathcal{M}$ and exogenous meteorological covariates are known. Let $\mathbf{X} \in \mathbb{R}^{(T+H) \times d}$ denote $d$ meteorological covariates over a context window of length $T$ and a forecast horizon of length $H$, and $\mathbf{y} \in \mathbb{R}^T$ the corresponding daily PV yield. Under standard supervised forecasting, a model predicts a horizon of length $H$ by conditioning on the target history, exogenous drivers over both past and forecast horizons, and plant information:

$$f_\theta : \left( \mathbf{y}_{1:T}, \mathbf{X}_{1:T}, \mathbf{X}_{T+1:T+H}, \mathcal{M} \right) \mapsto \hat{\mathbf{y}}_{T+1:T+H} \quad (1)$$

In the benchmark, the horizon covariates $\mathbf{X}_{T+1:T+H}$ are treated as known at evaluation time. This controlled setting allows forecasting models to be compared under the same future meteorological information, without confounding the results with errors from weather prediction. In the cold-start regime, however, the target history $\mathbf{y}_{1:T}$ is unavailable, rendering Eq. (1) inapplicable at inference time. The forecasting interface is retained by replacing the missing target history with a synthetic surrogate generated from plant metadata and weather covariates:

$$\tilde{\mathbf{y}}_{1:T} = g(\mathcal{M}, \mathbf{X}_{1:T}). \quad (2)$$

The generated sequence $\tilde{\mathbf{y}}$ is not treated as measured output, but as a structured proxy for the missing target history. It supplies the temporal context required by zero-shot foundation models and, where applicable, the synthetic target history used by classical reference forecasters.

### 2.2. Synthetic history generation

The initial stage of the process generates a daily synthetic PV yield history for each target system, which is then used as the initial context when measured production is unavailable. Two synthetic-history sources are considered in this study. The first of these is PVGIS, a widely utilised reference for estimating photovoltaic (PV) production from location, system configuration, and meteorological or satellite-derived information. The second source is OPAQUE, introduced in this work as an open, physics-based, satellite-free alternative with explicit and controllable modelling assumptions. For each synthetic-history source $s$, the unavailable measured target history $\mathbf{y}_{1:T}$ is replaced by a synthetic sequence of the same length:

$$\tilde{\mathbf{y}}_{1:T}^{(s)} = g_s(\mathcal{M}, \mathbf{X}_{1:T}), \quad s \in \{\text{PVGIS}, \text{OPAQUE}\}. \quad (3)$$

where $T$ is the synthetic-history length, $\mathcal{M}$ the plant metadata, $\mathbf{X}_{1:T}$ the weather covariates, and $\tilde{\mathbf{y}}_{1:T}^{(s)}$ the generated daily yield. The purpose of this stage is not exact reconstruction, but to provide a plausible temporal context capturing seasonality, capacity-normalised scale, and weather-driven variability. Comparing PVGIS and OPAQUE allows us to test whether downstream performance depends on the specific generator or mainly on the availability of a plausible synthetic context. The full OPAQUE formulation and the fidelity comparison with PVGIS are reported in Appendices Section A.1 and Section A.2.

### 2.3. Models

The benchmark comprises persistence baselines, a classical covariate-aware Prophet model (Taylor & Letham, 2018), and five zero-shot time-series foundation models: Chronos-2 (Ansari et al., 2025), Moirai 2.0 (Liu et al., 2025), TimesFM 2.5 (Das et al., 2024), TiRex (Auer et al., 2025), and TabPFN-TS (Hoo et al., 2025) (Table 1). The persistence-based baselines (naive and seasonal-naive) serve as covariate-free references that rely exclusively on the target series. Prophet adopts a classical decomposable formulation, fitting a piecewise-linear trend, Fourier-based seasonal components, and a linear regression on weather covariates to each series. The foundation models are evaluated on their published checkpoints in a strict zero-shot regime, with no per-site training or parameter updates; adaptation occurs exclusively through inference-time conditioning on the provided temporal context. The forecasters differ primarily in their treatment of covariates. TiRex and Moirai 2.0 remain univariate with respect to covariates, such that predictions

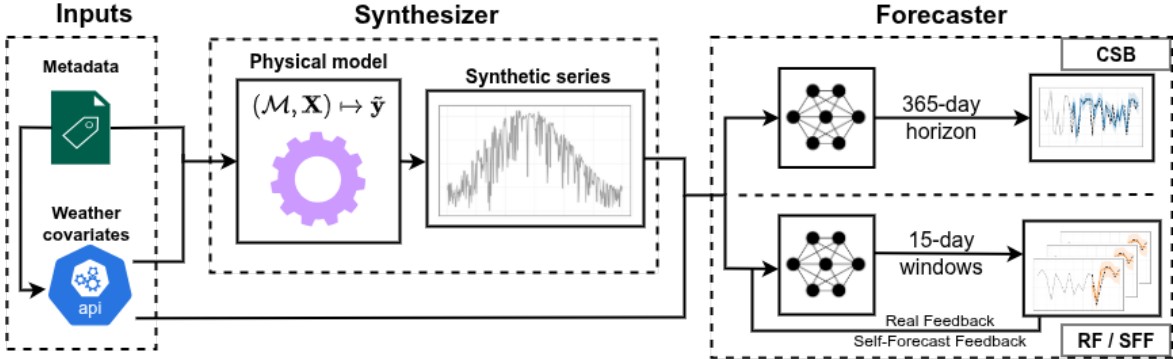

*Figure 1.* Overview of the proposed cold-start forecasting pipeline. Metadata and meteorological covariates are converted into a synthetic production history, which is used as a temporal context by the forecaster under CSB, RF, and SFF strategies.

depend solely on the target. In contrast, TimesFM 2.5 combines a pretrained univariate transformer backbone with an auxiliary linear regressor on the same covariates, estimated at inference time using the context window. Chronos-2 and TabPFN-TS instead model target and covariates jointly within a unified computation, leveraging pretraining on multivariate panels; notably, TabPFN-TS is the only method that additionally incorporates static plant metadata (e.g., peak power, tilt, azimuth, technology). Architectural details and covariate pathways are deferred to Appendix Section B.

*Table 1.* Forecaster zoo.

| Model | Family | Covariates |
|---|---|---|
| naive | persistence | univariate |
| seasonal-naive | persistence | univariate |
| Prophet | classical | past + future |
| Chronos-2 | zero-shot FM | past + future |
| Moirai 2.0 | zero-shot FM | univariate |
| TimesFM 2.5 | zero-shot FM | past + future |
| TiRex | zero-shot FM | univariate |
| TabPFN-TS | zero-shot FM | past + future + static |

### 2.4. Context strategies

The forecasting stage is evaluated under three scenarios: one 365-day forecast over the held-out year and two 15-day rolling-origin protocols. Let $\tau = 15$ denote the rolling forecast window. For the rolling protocols, the $k$-th forecast origin is defined as $t_k = T + k\tau$, where $T$ is the end of the synthetic-history context. Thus, $T + 1$ is the first day of the held-out evaluation year. The scenarios differ only in how the target-series context is populated. In all scenarios, the initial context is the synthetic history $\tilde{\mathbf{y}}_{1:T}^{(s)}$, which replaces the unavailable measured target history $\mathbf{y}_{1:T}$. In the rolling-origin protocols, this fixed synthetic history is extended at each origin $t_k$ with either measured production $\mathbf{y}_{T+1:t_k}$ or model-generated production $\hat{\mathbf{y}}_{T+1:t_k}$. The exogenous inputs are constructed consistently across scenarios. Representative traces are provided in Appendix D.

- **Cold-Start Baseline (CSB).** The target-series context is limited to the synthetic history $\tilde{\mathbf{y}}_{1:T}^{(s)}$. No measured target-

site production is provided. The model predicts the full held-out year $\hat{\mathbf{y}}_{T+1:T+365}$ in a single forward pass, yielding the strict synthetic-only cold-start setting.
- **Real Feedback (RF).** The target-series context is updated with measured production as it becomes available after commissioning. At origin $t_k$, the context is $[\tilde{\mathbf{y}}_{1:T}^{(s)}, \mathbf{y}_{T+1:t_k}]$, and the model predicts the next $\tau = 15$ days. The synthetic segment remains fixed, while the real segment grows across rolling origins.
- **Self-Forecast Feedback (SFF).** The target-series context is updated with the model's own previous forecasts instead of measured production. At origin $t_k$, the context is $[\tilde{\mathbf{y}}_{1:T}^{(s)}, \hat{\mathbf{y}}_{T+1:t_k}]$, and the model predicts the next $\tau = 15$ days. The synthetic segment remains fixed, while the predicted segment grows across rolling origins.

### 2.5. Datasets

The experimental evaluation is conducted on four publicly available rooftop-PV datasets spanning three continents and diverse climatic conditions. The benchmark includes 440 sites in total: 300 from Ausgrid, 10 from DKASC, 100 from UK-PV, and 30 from PVDAQ. For each site, the most recent full year of measured production is reserved for cold-start evaluation. Ausgrid (Ausgrid, 2014; Haessig, 2016) provides a substantial residential cohort in the Sydney area. DKASC (Desert Knowledge Australia Centre, 2024) contributes a smaller desert benchmark from Alice Springs. UK-PV (UK Power Networks, 2014) represents a temperate and cloudier European setting, while PVDAQ (National Renewable Energy Laboratory, 2023) extends the evaluation to multiple US climate zones.

### 2.6. Metrics

The evaluation targets the downstream forecasting performance of the considered models. Targets and forecasts are normalised by plant capacity before any metric is evaluated, $\mathbf{y}_t = \mathbf{y}_t^{(abs)}/P_{\text{peak}}$ and $\hat{\mathbf{y}}_t = \hat{\mathbf{y}}_t^{(abs)}/P_{\text{peak}}$, where $P_{\text{peak}}$ denotes the plant nominal peak capacity, both in

$\mathrm{kW\,h\,kWp^{-1}\,d^{-1}}$(specific-yield convention), so residential and commercial systems contribute to cross-cohort aggregates on equal footing. Evaluation is performed on the capacity-normalised series using the same three error metrics:

- **MAE**, the mean absolute error in $\mathrm{kW\,h\,kWp^{-1}\,d^{-1}}$;
- **RMSE**, in the same units, which penalises large deviations more heavily;
- **WAPE**, dimensionless and scale-free by construction, which remains stable at low production levels where MAPE becomes unreliable.

## 3. Experiments and Results

The aggregate results are reported in Table 2 and Table 3. Both tables show population-mean errors across all sites, models, context strategies, and synthetic-history sources. Extended results, per-cohort breakdowns and daily traces, are deferred to Sections C.1, C.2 and D.

*Table 2.* Cold-start MAE $(\mathrm{kW\,h\,kWp^{-1}\,d^{-1}})$. RF: *Real feedback*; SFF: *Self-Forecast Feedback*; CSB: *Cold-start baseline*.

| Model | OPAQUE context | | | PVGIS context | | |
|---|---|---|---|---|---|---|
| | CSB | RF | SFF | CSB | RF | SFF |
| naive | 1.630 | 1.425 | 1.639 | 1.674 | 1.424 | 1.684 |
| seasonal-naive | 1.606 | 1.629 | 1.615 | 1.510 | 1.531 | 1.518 |
| Prophet | **1.006** | 1.007 | 1.007 | 0.928 | 0.930 | 0.930 |
| TiRex | 1.356 | 1.061 | 1.598 | 1.280 | 1.053 | 1.594 |
| Moirai 2.0 | 1.297 | 1.109 | 1.318 | 1.280 | 1.108 | 1.316 |
| TimesFM 2.5 | 1.056 | 0.599 | 1.063 | 0.948 | 0.587 | 0.915 |
| Chronos-2 | 1.019 | 0.537 | **0.985** | **0.907** | 0.535 | **0.908** |
| TabPFN-TS | 1.016 | **0.514** | 1.025 | 0.923 | **0.512** | 0.937 |

*Table 3.* Cold-start RMSE $(\mathrm{kW\,h\,kWp^{-1}\,d^{-1}})$; abbreviations as in Table 2.

| Model | OPAQUE context | | | PVGIS context | | |
|---|---|---|---|---|---|---|
| | CSB | RF | SFF | CSB | RF | SFF |
| naive | 2.023 | 1.859 | 2.030 | 2.070 | 1.857 | 2.079 |
| seasonal-naive | 1.998 | 2.023 | 2.006 | 1.891 | 1.913 | 1.899 |
| Prophet | **1.221** | 1.222 | 1.222 | 1.137 | 1.138 | 1.138 |
| TiRex | 1.683 | 1.393 | 1.972 | 1.597 | 1.389 | 1.970 |
| Moirai 2.0 | 1.627 | 1.426 | 1.625 | 1.606 | 1.422 | 1.634 |
| TimesFM 2.5 | 1.273 | 0.793 | 1.282 | 1.159 | 0.779 | 1.123 |
| Chronos-2 | 1.244 | 0.737 | **1.203** | **1.114** | 0.735 | **1.113** |
| TabPFN-TS | 1.238 | **0.721** | 1.247 | 1.131 | **0.717** | 1.146 |

### 3.1. Cold-Start Baseline

In the *CSB*, no contextual observations are provided, resulting in a strictly zero-shot inference setting without conditioning signal. Under these conditions, performance differences between foundation models narrow, with all models clustering within approximately 0.05 MAE, indicating an intrinsic performance ceiling. On OPAQUE, Prophet remains marginally best (MAE 1.006, RMSE 1.221), with TabPFN-TS and Chronos-2 within 2 %. On PVGIS, Chronos-2 leads (MAE 0.907, RMSE 1.114). Errors approximately double

relative to *RF*, highlighting the limitation of zero-context inference. Two distinct mechanisms underlie this ceiling: Prophet produces a policy-invariant trajectory that is identical across regimes due to per-series fitting, whereas TabPFN-TS leverages static covariates ($\mathcal{M}$) to anchor predictions to plant characteristics even in the absence of informative temporal context.

### 3.2. Real Feedback

Under *RF*, foundation models (TabPFN-TS, Chronos-2, TimesFM 2.5) consistently outperform all baselines, achieving approximately 1.7–2× lower error than Prophet across both contexts and metrics. Despite operating in a zero-shot setting, these models effectively leverage the limited observed context available at inference time, indicating strong adaptability of pretrained temporal representations. Moirai 2.0 and TiRex remain less competitive in this regime, with errors roughly twice those of the leading three. Prophet remains largely invariant across regimes, consistent with its non-adaptive formulation.

### 3.3. Self-Forecast Feedback

Under *SFF*, Chronos-2 achieves the best performance across all metrics and contexts, with a clear margin over competing models. This setting corresponds to a fully zero-shot, autoregressive scenario in which no real observations are available and predictions are recursively fed back as context. The widening performance gap indicates superior robustness of Chronos-2 under distribution shift induced by self-conditioning. The same behaviour is reflected in the daily traces: TiRex rapidly drifts away from the measured envelope due to the absence of horizon-side meteorological covariates, whereas Chronos-2 and Moirai 2.0 preserve the seasonal structure of the signal.

## 4. Conclusions and Future Work

Cold-start PV forecasting is addressed via a zero-shot pipeline combining foundation models with a physics-based synthetic context. Across 440 sites, TabPFN-TS achieves the lowest error under *RF* (MAE 0.514, RMSE 0.721), while Chronos-2 is most robust under *SFF*. Covariate-aware foundation models consistently outperform classical baselines by approximately 1.7–2×. Performance is largely insensitive to the synthetic-context source: replacing OPAQUE with PVGIS yields negligible MAE/RMSE variation and unchanged ranking, indicating that performance is driven by the availability of plausible temporal context rather than the generator. Limitations include daily resolution, single-seed evaluation, and perfect-foresight ERA5 inputs, leading to optimistic estimates. Future work will investigate synthetic-data-driven fine-tuning, focusing on the trade-off between zero-shot generalisation, adaptation, and generator-induced bias.

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

## A. Synthetic Data Generator: OPAQUE Physics and Fidelity

This appendix specifies and validates the deterministic physics-based generator $g$ of Equation (2), which produces the synthetic histories consumed by the downstream forecasters. Section A.1 gives the full conversion chain from ERA5 meteorological covariates and commissioning-time plant metadata to a daily specific-yield sequence; Section A.2 reports a fidelity check of OPAQUE and PVGIS against measured ground truth on the four evaluation cohorts.

### A.1. OPAQUE physics

OPAQUE consumes commissioning-time plant metadata together with ERA5 reanalysis covariates retrieved from Open-Meteo, and emits a daily specific-yield sequence through three stages (irradiance transposition, thermal derating, and DC–AC integration) illustrated in Figure 2.

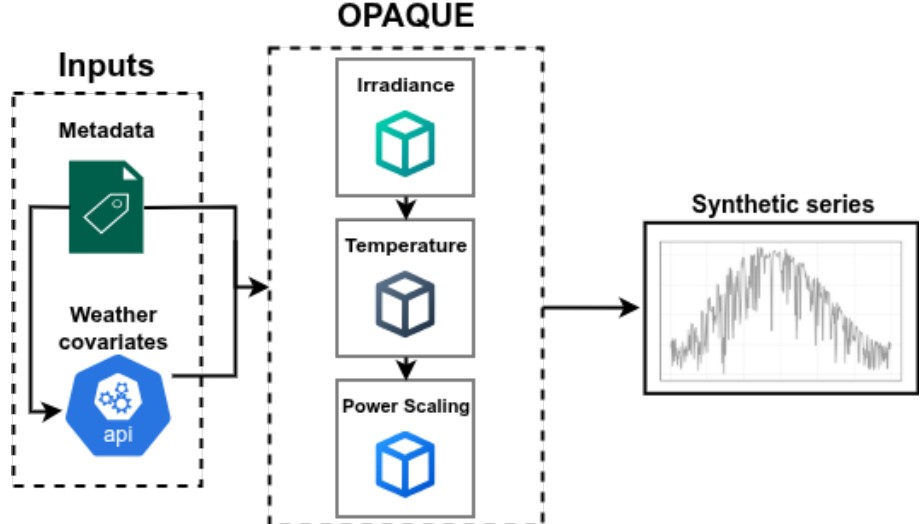

*Figure 2.* OPAQUE synthetic-history generator.

OPAQUE (Open Physics-based Acquisition of Quantitative Energy) is the deterministic physics-based generator that instantiates $g$ in Eq. (2), mapping commissioning-time information to a synthetic daily PV history suitable for cold-start forecasting. OPAQUE operates on two sources of information. The first consists of ERA5-based meteorological covariates (Hersbach et al., 2020) retrieved through Open-Meteo (Open-Meteo) at hourly resolution: global horizontal irradiance $G$ together with its ERA5-native beam and diffuse components $B$ and $D$, ambient temperature $T_a$, and wind speed $w$ at $10\,\mathrm{m}$. Downstream forecasters do not ingest the hourly stream but the daily aggregates derived from it, namely integrated irradiance $\sum_h G_h$, mean and maximum ambient temperature $\bar{T}_a, T_a^{\mathrm{max}}$, and maximum wind speed $w^{\mathrm{max}}$, together with daylight duration. The second consists of plant metadata $\mathcal{M}$ available at commissioning time, including peak power $P_{\mathrm{peak}}$, tilt $\beta$, azimuth $A$, thermal coefficient $\gamma_{\mathrm{Pmax}}$ (negative for crystalline-silicon modules), nominal operating cell temperature (NOCT), inverter efficiency $\eta_{\mathrm{inv}}$, tracking type, installation year, and disaggregated loss factors $(\ell_s, \ell_m, \ell_{dc}, \ell_{ac}, \ell_{sh})$ for soiling, module mismatch, DC-side and AC-side cabling, and shading, respectively. When site-specific values are unavailable, per-dataset literature-based priors are applied for soiling, module mismatch, and cabling, yielding a combined non-thermal multiplicative loss factor $\prod_i (1-\ell_i)\,\eta_{\mathrm{inv}}$ in the range 0.85–0.89 across the four evaluation cohorts (15.1 %, 13.3 %, 12.4 %, and 11.0 % total non-thermal loss for Ausgrid, DKASC, PVDAQ, and UK–PV, respectively, reflecting the local soiling and mismatch climate). Together with the typical NOCT-driven thermal derating ($\sim$2.5 % under summer conditions), these per-cohort priors bracket the fixed PVGIS 14 % aggregate loss prior, OPAQUE matches it on the cohort average rather than imposing it uniformly site-wise. OPAQUE constructs the daily synthetic yield through three stages (Fig. 2):

1. **Irradiance transposition** (hourly). The plane-of-array irradiance $I_{\mathrm{POA},h}$ is the sum of three components: a beam term obtained by geometric projection of $B_h$ via the angle-of-incidence ratio $\cos(\theta_h)/\sin(\alpha_h)$ (with $\theta_h$ the angle between the sun vector and the panel normal, and $\alpha_h$ the solar elevation), a sky-diffuse term obtained from $D_h$ via the Hay–Davies anisotropic model (Hay & Davies, 1980) (isotropic plus circumsolar contributions), and a ground-reflection term $G_h\,\rho\,(1-\cos\beta)/2$ with constant albedo $\rho = 0.2$. Solar position is computed by $\mathtt{pvlib}$ from $(\mathrm{lat}, \mathrm{lon}, \mathrm{datetime}_{\mathrm{UTC}})$;

transposition is implemented from first principles in NUMPY.

2. **Cell temperature and thermal derating** (hourly). A NOCT model (Ross, 1976) gives the cell temperature

$$T_{\text{cell},h} = T_{a,h} + \frac{\text{NOCT} - 20}{800} I_{\text{POA},h}, \tag{4}$$

with $I_{\text{POA},h}$ in $\text{W/m}^2$, temperatures in $^\circ\text{C}$, and $800\,\text{W/m}^2$ the NOCT reference irradiance. When $10\,\text{m}$ wind speed is available, the thermal-rise term is reduced by a small wind-cooling factor (about $5\,\%$ per $\text{m/s}$ above $1\,\text{m/s}$). The thermal derating factor follows the standard linear form (Skoplaki & Palyvos, 2009):

$$\eta_{T,h} = 1 + \gamma_{\text{Pmax}}\big(T_{\text{cell},h} - 25\big). \tag{5}$$

Since $\gamma_{\text{Pmax}} < 0$ and $25\,^\circ\text{C}$ is the STC reference temperature, $\eta_{T,h} < 1$ whenever $T_{\text{cell},h} > 25\,^\circ\text{C}$. The factor is clipped to the physically plausible range $[0.5,\ 1.1]$ to guard against extreme cell-temperature outliers.

3. **Daily DC-AC integration.** Following the PVWatts system model (Dobos, 2014), the hourly DC output is scaled to AC through inverter efficiency, plant-level losses, and module aging, and summed over the day:

$$
\begin{aligned}
\tilde{y}_t \;=\; & \sum_{h \in \text{day}(t)} \Delta h\, P_{\text{peak}}\, \frac{I_{\text{POA},h}}{G_{\text{STC}}}\, \eta_{T,h} \\
& \times \prod_i (1-\ell_i)\, \eta_{\text{inv}}\, (1-\delta)^{a_t},
\end{aligned}
\tag{6}
$$

where $\Delta h = 1\,\text{h}$ and the sum spans the $24\,\text{h}$ of day $t$; $G_{\text{STC}} = 1000\,\text{W/m}^2$, $\delta = 0.005$ is the annual module degradation rate (Jordan & Kurtz, 2013), and $a_t$ is the age of the plant in years at day $t$. When daily snowfall is available from the meteo aggregates, $\tilde{y}_t$ is further multiplied by a snow-cover penalty of $0.1$ if daily snowfall exceeds $5\,\text{mm}$, $0.5$ if it exceeds $1\,\text{mm}$, and $1$ otherwise.

Site identity and seasonal phase enter the pipeline through two complementary channels. First, the hourly Open-Meteo query is parameterised by the plant's coordinates $(\text{lat}, \text{lon})$, so $X_{1:L}$ inherits the local climatological seasonality of irradiance, ambient temperature, wind, and daylight duration for the specific location, including site-level cloud-cover variability rather than a smoothed climatology. Second, the solar position consumed by the Hay–Davies transposition is computed by pvlib as a function of $(\text{lat}, \text{lon}, \text{datetime}_{\text{UTC}})$, fixing the sun's declination and elevation curve at every hour of the year; the day-of-year dependence of the declination naturally reproduces the hemispheric asymmetry between summer and winter (e.g. a low-elevation sun in austral winter for southern-hemisphere sites such as DKASC and Ausgrid, and a low-elevation sun in boreal winter for UK-PV and PVDAQ). As a consequence, the synthetic history $\tilde{\mathbf{y}}_{<0}$ is generated independently for each plant $i$ from its own $(\mathcal{M}_i, X_i^{(\text{lat}_i, \text{lon}_i)})$ and its seasonal envelope reflects the actual geographic and astronomical conditions of that plant, not a templated profile shared across sites. By design, OPAQUE is open, datasheet-driven, and satellite-free: the full plant specification is explicitly exposed through $\mathcal{M}$, and the generator operates entirely on globally available ERA5 covariates, with solar position obtained from the open-source pvlib library and all remaining stages implemented from first principles, without reliance on any proprietary service.

## A.2. Synthetic-source fidelity

Table 4 reports the fidelity of OPAQUE and PVGIS against measured ground truth on the four evaluation cohorts; Figure 3 overlays both simulators against the cohort median-WAPE system. On Ausgrid and PVDAQ the two generators agree within 2.5 pp of WAPE. OPAQUE improves on DKASC by 2.5 pp, suggesting plant-specific losses are better captured than PVGIS's aggregate 14 % loss prior under clear-sky conditions; the larger UK-PV gap (9.7 pp) reflects PVGIS's SARAH3 radiation backend over Europe rather than the downstream PV chain.

*Table 4.* Fidelity of OPAQUE and PVGIS against measured production, per cohort. MAE, RMSE in $\mathrm{kW\,h\,kWp^{-1}\,d^{-1}}$.

|  |  | PVGIS | | | OPAQUE | | |  |
| Cohort | Sys. | WAPE | MAE | RMSE | WAPE | MAE | RMSE | $\Delta$WAPE (pp) |
| --- | --- | --- | --- | --- | --- | --- | --- | --- |
| Ausgrid | 300 | **25.4 %** | **0.92** | **1.11** | 28.0 % | 1.01 | 1.19 | −2.5 |
| DKASC | 10 | 13.1 % | 0.74 | 0.85 | **10.6 %** | **0.53** | **0.68** | 2.5 |
| UK-PV | 100 | **12.0 %** | **0.33** | **0.43** | 21.7 % | 0.60 | 0.78 | −9.7 |
| PVDAQ | 30 | **21.6 %** | **0.80** | **1.04** | 21.8 % | 0.82 | 1.07 | −0.2 |

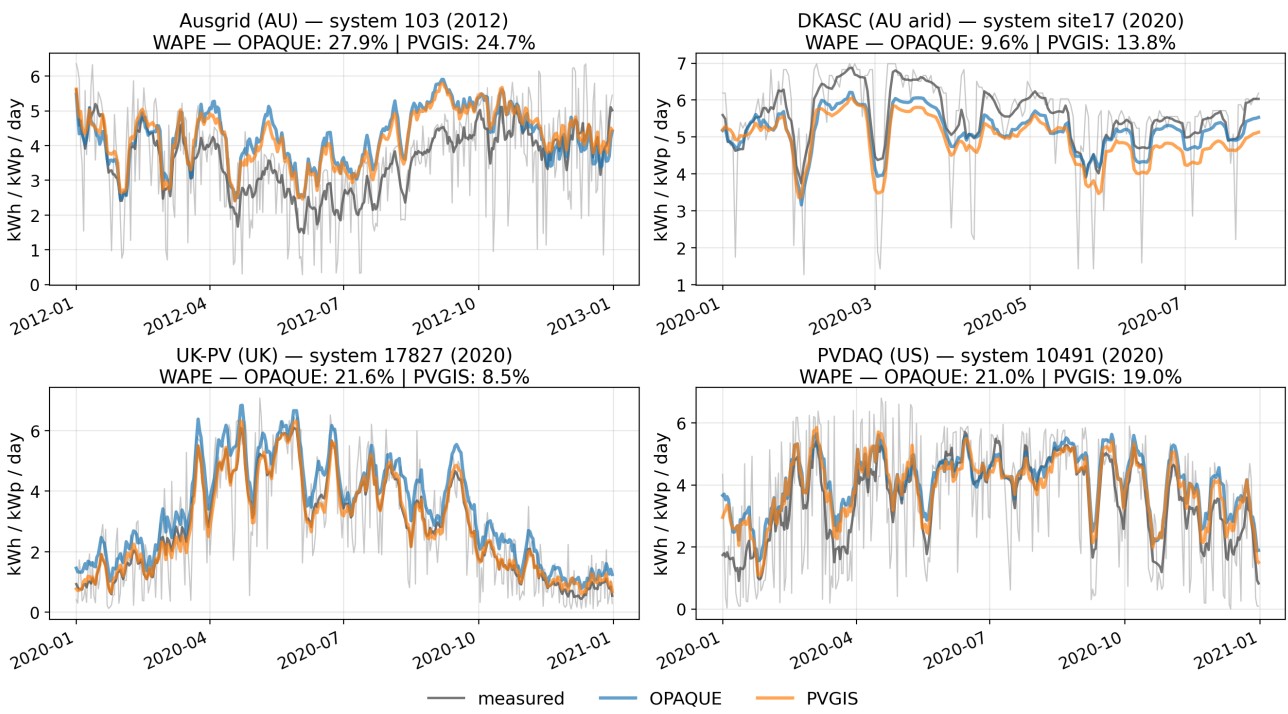

*Figure 3.* OPAQUE vs PVGIS fidelity against measured daily production, per-cohort median-WAPE system over the full evaluation year. Both simulators reproduce the seasonal envelope; OPAQUE matches PVGIS fidelity within a few percentage points of WAPE on every cohort while remaining satellite-free. The DKASC panel shows site 17 (2020), the same representative system displayed in the cold-start trace figures of Section D, and stops mid-year because no DKASC site in our cohort selection has a full 365-day window of measured ground truth; the OPAQUE and PVGIS curves are therefore truncated to the days where measured ground truth is available, and the reported WAPE is computed on that same support. To improve readability across an entire year of daily samples, the OPAQUE and PVGIS curves are shown as their centered 7-day rolling mean, which absorbs high-frequency cloudy/clear day fluctuations; the measured reference is shown both as the original daily series (faded gray) and as its 7-day rolling mean (darker overlay). All numerical fidelity metrics in this section are computed on the unsmoothed daily series.

# B. Forecaster Zoo: Architectural Details and Covariate Paths

This appendix complements Table 1 of the main text with a compact architectural overview of the forecasters in the benchmark, focusing on the architectural pathways through which the models consume dynamic exogenous covariates, with particular emphasis on the zero-shot foundation models. Rather than treating covariate support as a binary property, we distinguish three covariate-consumption regimes, labelled (M1)–(M3), which differ in how covariates enter the forecasting pipeline and in the class of dependencies they can represent. The taxonomy characterises the covariate pathway independently of the underlying model family and is used as the organising axis of the zoo. For full architectural details, the reader is referred to the original papers.

## B.1. Three regimes of covariate consumption

Covariates can affect a forecast either through explicit additive corrections or through joint non-linear interactions learned together with the target dynamics. In zero-shot forecasting, this distinction is particularly important because some foundation models are pretrained jointly on targets and covariates, whereas others remain strictly univariate and can only exploit covariates through an auxiliary covariate module at inference time. Together with models that ignore covariates entirely, this yields the three regimes considered here.

**Mode (M1), native multivariate.** The model is pretrained on multivariate panels in which target and covariates are jointly visible to the same loss. At inference time all channels are processed together by the same backbone, allowing the predictor to represent non-linear and state-dependent cross-variate interactions learned during pretraining.

**Mode (M2), univariate backbone with external regressor.** The forecasting backbone remains strictly univariate, while covariates influence the prediction through a separate additive covariate pathway, typically parameterised by a linear regression fitted on paired covariate–target observations. The final prediction combines the backbone forecast with the covariate-dependent additive contribution. In this regime, covariate effects are not learned jointly within the forecasting backbone and are therefore limited by the expressiveness of the auxiliary covariate pathway.

**Mode (M3), univariate.** The forecast depends only on the past target history and on the inductive biases embedded in the model itself. Released zero-shot foundation models documented as univariate-only belong to this regime, as do the covariate-free reference baselines used for comparison.

The three regimes can therefore be ordered by the information admitted into the forecast: (M3) ignores covariates, (M2) incorporates them through an explicit auxiliary covariate module, and (M1) integrates them directly into the pretrained forecasting backbone, enabling learned cross-variate interactions.

## B.2. Compact summary of the zoo

Table 5 summarises the eight forecasters in terms of model family, forecasting mechanism, supported information sets, and covariate-consumption regime.

*Table 5.* Architectural and information-set summary of the forecaster zoo. Modes (M1)–(M3) correspond to the covariate-consumption regimes defined in Section B.1. Past and Future denote dynamic covariates aligned with the history window and forecast horizon, respectively, while Static indicates support for time-invariant metadata.

| Model | Family | Mechanism | Past | Future | Static | Mode |
|---|---|---|---|---|---|---|
| Chronos-2 | zero-shot FM | T5 encoder | ✓ | ✓ | ✗ | (M1) |
| Moirai 2.0 | zero-shot FM | decoder-only transformer | ✗ | ✗ | ✗ | (M3) |
| TimesFM 2.5 | zero-shot FM | decoder-only patch transformer | ✓ | ✓ | ✗ | (M2) |
| TiRex | zero-shot FM | xLSTM | ✗ | ✗ | ✗ | (M3) |
| TabPFN-TS | zero-shot FM | TabPFN-v2 with time features | ✓ | ✓ | ✓ | (M1) |

## C. Cold-Start Benchmark: Aggregate and Per-Cohort Tables

This appendix collects every numerical result of the cold-start benchmark across the two synthetic contexts (OPAQUE and PVGIS). All tables and discussion below use the three context strategies of Section 2.4, abbreviated as *Real Feedback* (RF), *Self-Forecast Feedback* (SFF), and *Cold-Start Baseline* (CSB). Section C.1 reports population-level summaries on the full 440-site grid; Section C.2 decomposes the same numbers by evaluation cohort.

### C.1. Aggregate cross-context benchmark

This section complements the MAE and RMSE tables of Section 3 with the dimensionless WAPE values and the signed penalties for both synthetic contexts. Table 6 reports overall WAPE under RF, SFF, and CSB for OPAQUE and PVGIS side-by-side; Table 7 reports the autoregressive penalty $\Delta_{\mathrm{SFF}} = \mathrm{WAPE}_{\mathrm{SFF}} - \mathrm{WAPE}_{\mathrm{RF}}$ and the cold-start penalty $\Delta_{\mathrm{CSB}} = \mathrm{WAPE}_{\mathrm{CSB}} - \mathrm{WAPE}_{\mathrm{RF}}$.

*Table 6.* Overall WAPE (%) under RF, SFF, and CSB.

| Model | OPAQUE context | | | PVGIS context | | |
|---|---|---|---|---|---|---|
| | RF | SFF | CSB | RF | SFF | CSB |
| naive | 42.65 | 50.74 | 50.50 | 42.64 | 52.36 | 52.10 |
| seasonal-naive | 49.96 | 49.46 | 49.15 | 47.02 | 46.56 | 46.30 |
| Prophet | 31.40 | 31.40 | **31.30** | 28.93 | 28.93 | 28.81 |
| TiRex | 31.97 | 49.77 | 41.80 | 31.73 | 49.46 | 39.36 |
| Moirai 2.0 | 33.33 | 40.13 | 39.74 | 33.32 | 40.12 | 39.11 |
| TimesFM 2.5 | 18.11 | 33.09 | 32.77 | 17.72 | 28.47 | 29.43 |
| Chronos-2 | 16.14 | **30.67** | 31.64 | 16.08 | **28.24** | **28.14** |
| TabPFN-TS | **15.46** | 31.91 | 31.55 | **15.39** | 29.14 | 28.58 |

*Table 7.* Context-strategy penalties on WAPE in pp: $\Delta_{\mathrm{SFF}}$ (SFF vs. RF) and $\Delta_{\mathrm{CSB}}$ (CSB vs. RF), OPAQUE and PVGIS contexts.

| Model | OPAQUE | | PVGIS | |
|---|---|---|---|---|
| | $\Delta_{\mathrm{SFF}}$ | $\Delta_{\mathrm{CSB}}$ | $\Delta_{\mathrm{SFF}}$ | $\Delta_{\mathrm{CSB}}$ |
| naive | 8.09 | 7.85 | 9.72 | 9.47 |
| seasonal-naive | −0.50 | −0.81 | −0.46 | −0.72 |
| Prophet | **0.00** | **−0.10** | **0.00** | **−0.12** |
| TiRex | 17.80 | 9.83 | 17.73 | 7.63 |
| Moirai 2.0 | 6.80 | 6.41 | 6.80 | 5.79 |
| TimesFM 2.5 | 14.98 | 14.65 | 10.75 | 11.71 |
| Chronos-2 | 14.53 | 15.50 | 12.16 | 12.06 |
| TabPFN-TS | 16.45 | 16.09 | 13.76 | 13.19 |

From Table 6, every TSFM's RF WAPE differs by less than $0.5\,\mathrm{pp}$ between OPAQUE and PVGIS, and the relative ordering of the three leaders (TabPFN-TS, Chronos-2, TimesFM 2.5) is preserved under RF and SFF in both contexts, confirming that the headline of Section 3 is not an artefact of the specific synthetic source. Under the strict-annual CSB the leadership shifts: on OPAQUE Prophet reaches the lowest WAPE ($31.30\,\%$, ahead of TabPFN-TS at $31.55\,\%$ and Chronos-2 at $31.64\,\%$) by virtue of its strategy-invariance, while on PVGIS Chronos-2 leads ($28.14\,\%$). The penalty view of Table 7 shows that Moirai 2.0 records the smallest absolute foundation model penalties in both contexts ($\Delta_{\mathrm{SFF}} = 6.80\,\mathrm{pp}$, $\Delta_{\mathrm{CSB}} = 6.41\,\mathrm{pp}$ in OPAQUE; $\Delta_{\mathrm{SFF}} = 6.80\,\mathrm{pp}$, $\Delta_{\mathrm{CSB}} = 5.79\,\mathrm{pp}$ in PVGIS), indicating high robustness to the context-distribution shift even though its absolute WAPE remains above the top tier; the three leading foundation models cluster in a narrower CSB band but pay roughly twice the penalty.

## C.2. Per-cohort breakdown

This section decomposes the aggregate benchmark of Section C.1 by evaluation cohort. Two parallel sets of three tables are reported, one per synthetic context and one per metric: WAPE, MAE, RMSE under OPAQUE in Tables 8 to 10, and the corresponding three under PVGIS in Tables 11 to 13. All six tables are on the same grid of 8 forecasters × 3 context strategies (RF, SFF, CSB) × 4 cohorts; cohorts are ordered by their irradiance-variability gradient (DKASC, PVDAQ, Ausgrid, UK-PV), from arid clear-sky to temperate-cloudy.

### C.2.1. OPAQUE CONTEXT

*Table 8.* Per-cohort WAPE (%), OPAQUE context.

| Model | DKASC | | | PVDAQ | | | Ausgrid | | | UK-PV | | |
|---|---|---|---|---|---|---|---|---|---|---|---|---|
| | RF | SFF | CSB | RF | SFF | CSB | RF | SFF | CSB | RF | SFF | CSB |
| naive | 18.46 | 25.12 | 25.12 | 34.44 | 53.28 | 53.51 | 41.46 | 44.20 | 43.79 | 51.10 | 72.11 | 72.20 |
| seasonal-naive | 19.79 | 19.79 | 19.79 | 46.91 | 46.07 | 46.26 | 50.28 | 50.28 | 49.76 | 52.93 | 50.98 | 51.11 |
| Prophet | 12.96 | 12.96 | 12.96 | 31.93 | 31.93 | **32.14** | 34.31 | 34.31 | **34.08** | 24.39 | 24.39 | 24.56 |
| TiRex | 14.74 | 18.72 | 18.09 | 26.22 | 49.10 | 40.19 | 30.48 | 44.26 | 42.07 | 39.87 | 69.54 | 43.84 |
| Moirai 2.0 | 15.60 | 18.32 | 18.48 | 27.59 | 41.27 | 41.21 | 32.12 | 38.50 | 37.62 | 40.44 | 46.84 | 47.77 |
| TimesFM 2.5 | 9.99 | 12.92 | 12.81 | 16.38 | 32.80 | 32.78 | 18.12 | 36.74 | 36.34 | 19.42 | 24.30 | 24.08 |
| Chronos-2 | 9.76 | 12.99 | 12.95 | 14.43 | **31.62** | 32.53 | 16.09 | **33.95** | 34.93 | 17.45 | **22.36** | 23.42 |
| TabPFN-TS | **9.62** | **12.58** | **12.57** | **13.88** | 32.47 | 32.66 | **15.47** | 35.31 | 34.85 | **16.48** | 23.51 | **23.23** |

*Table 9.* Per-cohort MAE ($\mathrm{kW\,h\,kWp^{-1}\,d^{-1}}$), OPAQUE context.

| Model | DKASC | | | PVDAQ | | | Ausgrid | | | UK-PV | | |
|---|---|---|---|---|---|---|---|---|---|---|---|---|
| | RF | SFF | CSB | RF | SFF | CSB | RF | SFF | CSB | RF | SFF | CSB |
| naive | 0.989 | 1.306 | 1.306 | 1.243 | 1.934 | 1.927 | 1.451 | 1.485 | 1.470 | 1.446 | 2.046 | 2.028 |
| seasonal-naive | 1.062 | 1.062 | 1.062 | 1.628 | 1.603 | 1.598 | 1.693 | 1.693 | 1.675 | 1.495 | 1.441 | 1.430 |
| Prophet | 0.700 | 0.700 | 0.700 | 1.086 | 1.086 | **1.086** | 1.116 | 1.116 | **1.104** | 0.690 | 0.690 | 0.688 |
| TiRex | 0.793 | 0.996 | 0.969 | 0.938 | 1.781 | 1.375 | 1.060 | 1.479 | 1.400 | 1.128 | 1.962 | 1.227 |
| Moirai 2.0 | 0.845 | 0.995 | 1.004 | 1.000 | 1.447 | 1.425 | 1.117 | 1.313 | 1.281 | 1.145 | 1.326 | 1.337 |
| TimesFM 2.5 | 0.544 | 0.698 | 0.691 | 0.584 | 1.118 | 1.113 | 0.619 | 1.195 | 1.178 | 0.549 | 0.687 | 0.674 |
| Chronos-2 | 0.530 | 0.699 | 0.696 | 0.515 | **1.082** | 1.102 | 0.554 | **1.103** | 1.132 | 0.493 | **0.632** | 0.655 |
| TabPFN-TS | **0.522** | **0.677** | **0.677** | **0.496** | 1.106 | 1.104 | **0.532** | 1.148 | 1.129 | **0.465** | 0.664 | **0.650** |

*Table 10.* Per-cohort RMSE ($\mathrm{kW\,h\,kWp^{-1}\,d^{-1}}$), OPAQUE context.

| Model | DKASC | | | PVDAQ | | | Ausgrid | | | UK-PV | | |
|---|---|---|---|---|---|---|---|---|---|---|---|---|
| | RF | SFF | CSB | RF | SFF | CSB | RF | SFF | CSB | RF | SFF | CSB |
| naive | 1.395 | 1.667 | 1.667 | 1.662 | 2.276 | 2.272 | 1.882 | 1.824 | 1.810 | 1.897 | 2.613 | 2.597 |
| seasonal-naive | 1.497 | 1.497 | 1.497 | 2.008 | 1.982 | 1.977 | 2.087 | 2.087 | 2.070 | 1.891 | 1.824 | 1.815 |
| Prophet | 0.915 | 0.915 | 0.915 | 1.305 | **1.305** | **1.304** | 1.346 | 1.346 | **1.336** | 0.854 | 0.854 | 0.852 |
| TiRex | 1.214 | 1.396 | 1.342 | 1.283 | 2.153 | 1.684 | 1.396 | 1.819 | 1.735 | 1.435 | 2.434 | 1.534 |
| Moirai 2.0 | 1.242 | 1.353 | 1.360 | 1.301 | 1.730 | 1.709 | 1.437 | 1.615 | 1.612 | 1.450 | 1.649 | 1.675 |
| TimesFM 2.5 | 0.816 | 0.915 | 0.914 | 0.809 | 1.328 | 1.328 | 0.824 | 1.435 | 1.416 | 0.696 | 0.849 | **0.831** |
| Chronos-2 | 0.803 | 0.914 | 0.913 | 0.759 | 1.316 | 1.340 | 0.760 | **1.330** | 1.370 | 0.655 | **0.815** | 0.840 |
| TabPFN-TS | **0.795** | **0.880** | **0.881** | **0.754** | 1.346 | 1.342 | **0.746** | 1.381 | 1.363 | **0.629** | 0.852 | 0.836 |

Under RF (Table 8), the three leading TSFMs rank consistently across cohorts, matching the irradiance-variability gradient: all three reach sub-10 % WAPE on the arid DKASC cohort (TabPFN-TS 9.62 %, Chronos-2 9.76 %, TimesFM 2.5 9.99 %)

and degrade by 7–10 pp on the cloudy UK-PV cohort. Under CSB, the DKASC floor stays around 13 % WAPE for all three top TSFMs, while PVDAQ and Ausgrid inflate by 12–22 pp; TabPFN-TS still leads on Ausgrid and UK-PV. Under SFF, Chronos-2 takes the cohort-level lead on three of four cohorts (DKASC, PVDAQ, Ausgrid), while Prophet's invariance pushes it ahead on UK-PV. Moirai 2.0 sits in an intermediate band, well above persistence on every cohort but still trailing the leading three by 5–10 pp; TiRex sits close to the persistence baselines beyond RF. The same leaderboard is preserved across the absolute-error metrics in Tables 9 and 10.

## C.2.2. PVGIS CONTEXT

*Table 11.* Per-cohort WAPE (%), PVGIS context.

| Model | DKASC | | | PVDAQ | | | Ausgrid | | | UK-PV | | |
| | RF | SFF | CSB | RF | SFF | CSB | RF | SFF | CSB | RF | SFF | CSB |
|---|---|---|---|---|---|---|---|---|---|---|---|---|
| naive | 16.41 | 25.28 | 25.28 | 34.37 | 51.34 | 51.57 | 41.42 | 45.28 | 44.84 | 51.13 | 76.29 | 76.40 |
| seasonal-naive | 20.39 | 20.39 | 20.39 | 45.80 | 44.97 | 45.15 | 46.05 | 46.05 | 45.62 | 52.72 | 50.93 | 51.04 |
| Prophet | 13.89 | 13.89 | 13.89 | 30.02 | 30.02 | **30.24** | 31.62 | 31.62 | 31.38 | 21.92 | 21.92 | 22.04 |
| TiRex | 13.85 | 19.87 | 18.80 | 26.30 | 49.72 | 38.42 | 30.30 | 45.27 | 39.66 | 39.24 | 64.57 | 40.58 |
| Moirai 2.0 | 15.24 | 19.04 | 19.17 | 27.54 | 39.81 | 39.70 | 31.98 | 38.33 | 37.57 | 40.68 | 47.44 | 45.36 |
| TimesFM 2.5 | 9.37 | 13.95 | 13.78 | 16.09 | 31.40 | 30.72 | 17.66 | **30.81** | 32.26 | 19.12 | 21.92 | 21.98 |
| Chronos-2 | **8.99** | 13.75 | **13.66** | 14.56 | **29.48** | 30.51 | 16.14 | 31.22 | **30.92** | 16.98 | **20.28** | 20.43 |
| TabPFN-TS | 9.06 | **13.62** | 13.70 | **14.03** | 30.73 | 30.94 | **15.45** | 32.14 | 31.87 | **16.17** | 21.12 | **19.38** |

*Table 12.* Per-cohort MAE (kW h kWp$^{-1}$ d$^{-1}$), PVGIS context.

| Model | DKASC | | | PVDAQ | | | Ausgrid | | | UK-PV | | |
| | RF | SFF | CSB | RF | SFF | CSB | RF | SFF | CSB | RF | SFF | CSB |
|---|---|---|---|---|---|---|---|---|---|---|---|---|
| naive | 0.903 | 1.379 | 1.379 | 1.242 | 1.866 | 1.860 | 1.450 | 1.514 | 1.498 | 1.447 | 2.166 | 2.147 |
| seasonal-naive | 1.130 | 1.130 | 1.130 | 1.598 | 1.573 | 1.569 | 1.550 | 1.550 | 1.535 | 1.491 | 1.440 | 1.429 |
| Prophet | 0.790 | 0.790 | 0.790 | 1.043 | 1.043 | **1.043** | 1.025 | 1.025 | 1.012 | 0.621 | 0.621 | 0.619 |
| TiRex | 0.763 | 1.094 | 1.047 | 0.941 | 1.811 | 1.329 | 1.053 | 1.510 | 1.319 | 1.111 | 1.827 | 1.137 |
| Moirai 2.0 | 0.852 | 1.071 | 1.078 | 1.001 | 1.410 | 1.386 | 1.112 | 1.304 | 1.278 | 1.152 | 1.344 | 1.271 |
| TimesFM 2.5 | 0.523 | 0.793 | 0.782 | 0.579 | 1.079 | 1.061 | 0.605 | **1.000** | 1.041 | 0.540 | 0.622 | 0.617 |
| Chronos-2 | **0.501** | 0.783 | 0.778 | 0.522 | **1.028** | 1.052 | 0.556 | 1.011 | **0.997** | 0.479 | **0.575** | 0.574 |
| TabPFN-TS | 0.504 | **0.772** | **0.777** | **0.502** | 1.064 | 1.063 | **0.532** | 1.042 | 1.028 | **0.457** | 0.598 | **0.544** |

*Table 13.* Per-cohort RMSE (kW h kWp$^{-1}$ d$^{-1}$), PVGIS context.

| Model | DKASC | | | PVDAQ | | | Ausgrid | | | UK-PV | | |
| | RF | SFF | CSB | RF | SFF | CSB | RF | SFF | CSB | RF | SFF | CSB |
|---|---|---|---|---|---|---|---|---|---|---|---|---|
| naive | 1.330 | 1.633 | 1.633 | 1.658 | 2.196 | 2.192 | 1.881 | 1.862 | 1.848 | 1.898 | 2.733 | 2.717 |
| seasonal-naive | 1.490 | 1.490 | 1.490 | 1.988 | 1.964 | 1.959 | 1.920 | 1.920 | 1.905 | 1.915 | 1.853 | 1.843 |
| Prophet | 0.949 | 0.949 | 0.949 | 1.276 | 1.276 | **1.276** | 1.246 | 1.246 | 1.234 | 0.792 | 0.792 | 0.789 |
| TiRex | 1.160 | 1.422 | 1.342 | 1.289 | 2.200 | 1.647 | 1.400 | 1.859 | 1.649 | 1.409 | 2.282 | 1.424 |
| Moirai 2.0 | 1.191 | 1.354 | 1.358 | 1.301 | 1.703 | 1.673 | 1.427 | 1.608 | 1.603 | 1.463 | 1.717 | 1.619 |
| TimesFM 2.5 | 0.757 | 0.945 | 0.940 | 0.805 | 1.301 | 1.291 | 0.809 | **1.223** | 1.267 | 0.688 | 0.789 | 0.781 |
| Chronos-2 | **0.740** | 0.935 | 0.931 | 0.768 | **1.269** | 1.298 | 0.763 | 1.223 | **1.211** | 0.641 | **0.752** | 0.753 |
| TabPFN-TS | 0.749 | **0.916** | **0.921** | **0.759** | 1.316 | 1.312 | **0.744** | 1.254 | 1.243 | **0.621** | 0.792 | **0.728** |

Under PVGIS context (Table 11) the per-cohort leaderboard remains substantially unchanged: TabPFN-TS, Chronos-2, and TimesFM 2.5 continue to occupy the three top positions on every cohort under RF (DKASC sub-10 % WAPE, UK-PV at

17–19 %), and TabPFN-TS retains the cohort-level lead on three out of four cohorts under CSB. Two systematic differences with respect to OPAQUE are visible in the absolute-error metrics of Tables 12 and 13: PVGIS yields slightly lower MAE/RMSE on UK-PV across all backbones (consistent with the SARAH3 satellite radiation product over Europe), and slightly higher errors on DKASC under RF for the three leading TSFMs (consistent with PVGIS's aggregate 14 % loss prior under clear-sky conditions, identified in Section A.2). Under SFF and CSB on UK-PV, Prophet's strategy-invariance translates into the smallest absolute errors on both metrics, the same phenomenon observed in OPAQUE context.

## D. Representative Daily Forecast Traces

This appendix complements aggregate metrics with full-year daily traces for one representative system per cohort. For each model, two figures are reported (OPAQUE and PVGIS), each showing a $2 \times 2$ grid over cohorts (Ausgrid, DKASC, UK-PV, PVDAQ) and the three context strategies of Section 2.4: *Real Feedback* (RF), *Self-Forecast Feedback* (SFF), and *Cold-Start Baseline* (CSB). RF, SFF, and CSB are used as abbreviations throughout the rest of this appendix.

**Interpretation.**

Following the context strategies defined in Section 2.4, all traces start from the same synthetic history $\tilde{\mathbf{y}}_{1:T}^{(s)}$, which replaces the unavailable measured target history at commissioning time. The three strategies differ in how this target-series context is used during the held-out evaluation year. In CSB, the context remains fixed as $\tilde{\mathbf{y}}_{1:T}^{(s)}$ and the model produces a single annual forecast $\hat{\mathbf{y}}_{T+1:T+365}$. In RF and SFF, forecasts are produced in rolling windows of length $\tau = 15$ from origins $t_k = T + k\tau$. RF extends the context with measured production $\mathbf{y}_{T+1:t_k}$, whereas SFF extends it with previously generated forecasts $\hat{\mathbf{y}}_{T+1:t_k}$. Thus, the synthetic segment is common to all strategies, while only the post-commissioning extension of the context differs.

---

**Algorithm 1** Rolling context update for RF and SFF

---

1: **Input:** synthetic history $\tilde{\mathbf{y}}_{1:T}^{(s)}$, rolling window $\tau = 15$, held-out horizon 365
2: Initialize:
$$\mathcal{C}_0^{\text{RF}} = \mathcal{C}_0^{\text{SFF}} = \tilde{\mathbf{y}}_{1:T}^{(s)}$$
3: **for** $k = 0, \ldots, \lceil 365/\tau \rceil - 1$ **do**
4:     Set $t_k = T + k\tau$
5:     For $r \in \{\text{RF}, \text{SFF}\}$, compute:
$$\hat{\mathbf{y}}_{t_k+1:t_k+\tau}^{r} = f_\theta \left( \mathcal{C}_k^{r}, \mathbf{X}_{1:t_k+\tau}, \mathcal{M} \right)$$
6:     Update the RF context with the measured block:
$$\mathcal{C}_{k+1}^{\text{RF}} = \left[ \mathcal{C}_k^{\text{RF}}, \mathbf{y}_{t_k+1:t_k+\tau} \right]$$
7:     Update the SFF context with the predicted block:
$$\mathcal{C}_{k+1}^{\text{SFF}} = \left[ \mathcal{C}_k^{\text{SFF}}, \hat{\mathbf{y}}_{t_k+1:t_k+\tau}^{\text{SFF}} \right]$$
8: **end for**

---

The algorithm makes explicit that RF and SFF share the same forecasting schedule, horizon covariates, and initial synthetic context. Their difference lies only in the information used to extend the target-series context after each origin. RF therefore represents a post-commissioning regime in which real telemetry progressively becomes available. SFF represents a stricter no-telemetry regime, where the model must remain stable under its own recursively generated context. CSB is stricter in a different sense: it does not update the context at all and evaluates whether a single synthetic-only context can support an annual forecast. This distinction is central to the interpretation of the traces. RF measures how quickly a model benefits from measured post-commissioning production. The two no-telemetry strategies probe complementary failure modes: CSB exposes long-horizon extrapolation drift, because the context is fixed and no autoregressive feedback is available; SFF exposes recursive autoregressive drift, because model biases can compound over successive rolling windows. In the no-telemetry regimes, the traces provide a qualitative indication of whether a model preserves the broad seasonal structure of measured production.

**Selection.** The representative site for each cohort is selected per model as the lower-median system by RF-WAPE, restricted to the sites whose number of measured eval-year days is at least 75 % of the largest number of measured eval-year days observed in that cohort; ties are broken lexicographically by system identifier. The coverage gate matters mostly for DKASC, where a few sites only have measurements for part of the eval year (sensor downtime or late data start). Without the gate, the median could land on one of those sites and its trace would appear cut off mid-year, misleadingly suggesting a model failure. On cohorts where every site has a full eval year (Ausgrid, UK-PV, PVDAQ in our data) the gate is inactive and selects the same site it would without it. RF is used as the anchor to ensure the same system is displayed across all three strategies and

both synthetic contexts within each model's figure. Different models may nonetheless select different representative sites, as RF-WAPE rankings vary by cohort.

**Reading and visualisation.** Each panel reports per-site WAPE in the title, which may differ from cohort averages. The horizontal axis spans the full eval-year calendar; the vertical axis is daily specific yield in $\mathrm{kW\,h\,kWp^{-1}\,d^{-1}}$. Curves are colour-coded consistently with the figure-level legend at the bottom: dark gray for the measured ground truth, blue for RF, green for SFF, and red for CSB. To suppress day-to-day weather noise without hiding the underlying signal, every line is rendered as a centered 7-day rolling mean drawn at full opacity; in addition, the measured series alone is shown as a faded daily-resolution background (low alpha, thin line) so that the noise envelope of the actual plant is visible while the four forecast curves remain readable. Daily-resolution traces are intentionally not drawn for the model predictions to avoid stacking four overlapping noise bands on the same panel. When two strategies coincide pairwise (e.g., RF $\equiv$ SFF for naive, or RF $\equiv$ SFF $\equiv$ CSB for Prophet), the duplicate curves are merged into a single line whose legend label uses the $\equiv$ symbol, and the WAPE annotation collapses accordingly. All reported metrics (WAPE in titles and in Section C) are computed on the unsmoothed daily series; the rolling mean is purely a display device and never enters the error calculation.

## D.1. Naive

Persistence baseline carrying the last observed daily yield through the 15-day horizon. Under RF the prediction is re-anchored every 15 days, producing a piecewise-constant trace that follows the seasonal envelope; under SFF and CSB the value is fixed at the last entry of the synthetic surrogate $\tilde{\mathbf{y}}_{<0}$ and the trace collapses to a year-long constant.

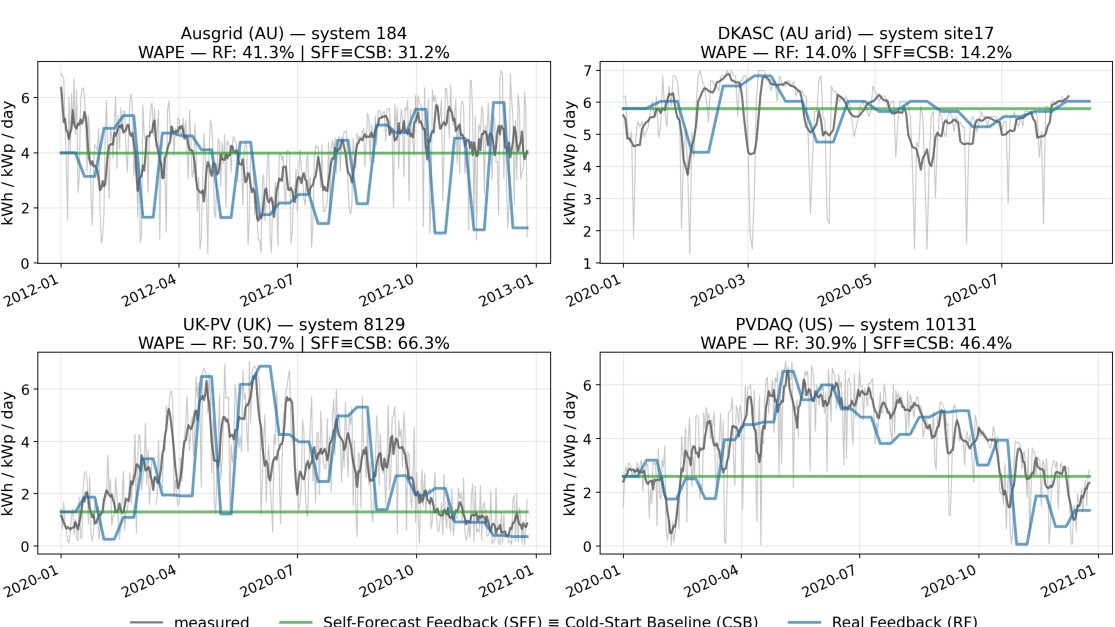

*Figure 4.* Naive baseline, OPAQUE synthetic context.

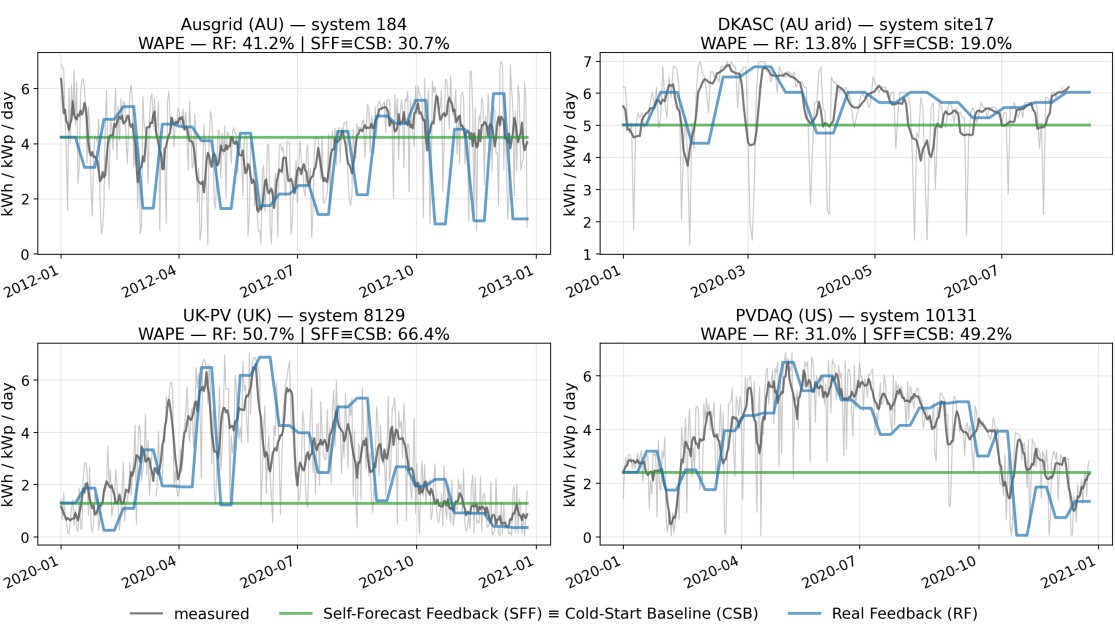

*Figure 5.* Naive baseline, PVGIS synthetic context.

## D.2. Seasonal-naive

Copies the same calendar day from the previous year. Under RF the previous year is the measured production and the trace inherits the day-to-day variability of the actual plant; under SFF and CSB only the synthetic surrogate is accessible, so the trace reproduces the simulator's seasonality (smoother than the real plant's; PVGIS is biased high on DKASC by its aggregate $14\%$ loss prior).

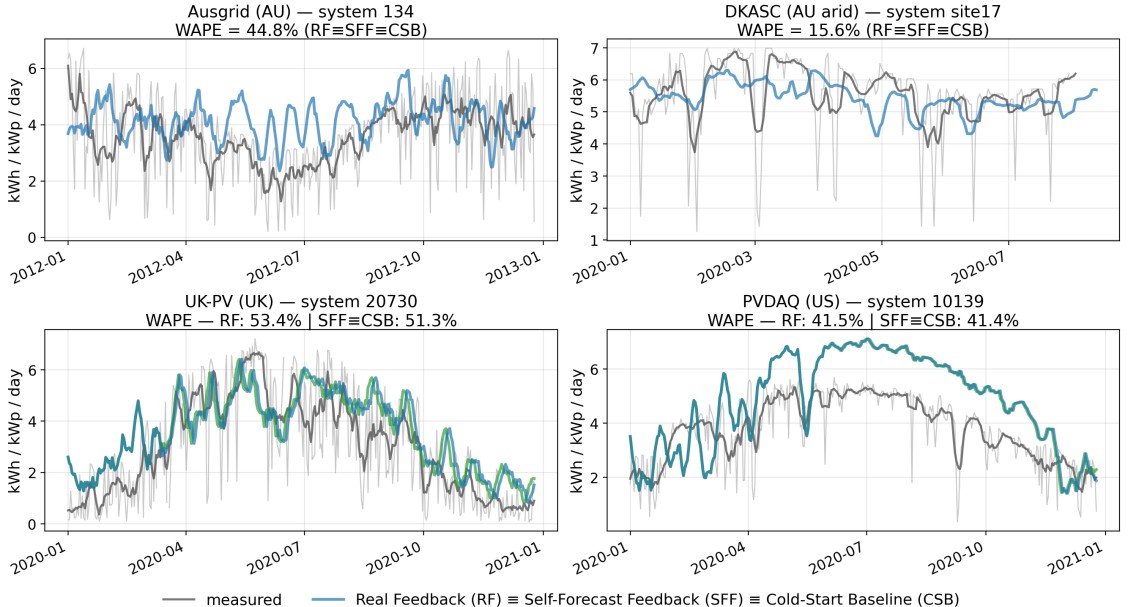

*Figure 6.* Seasonal-naive, OPAQUE synthetic context.

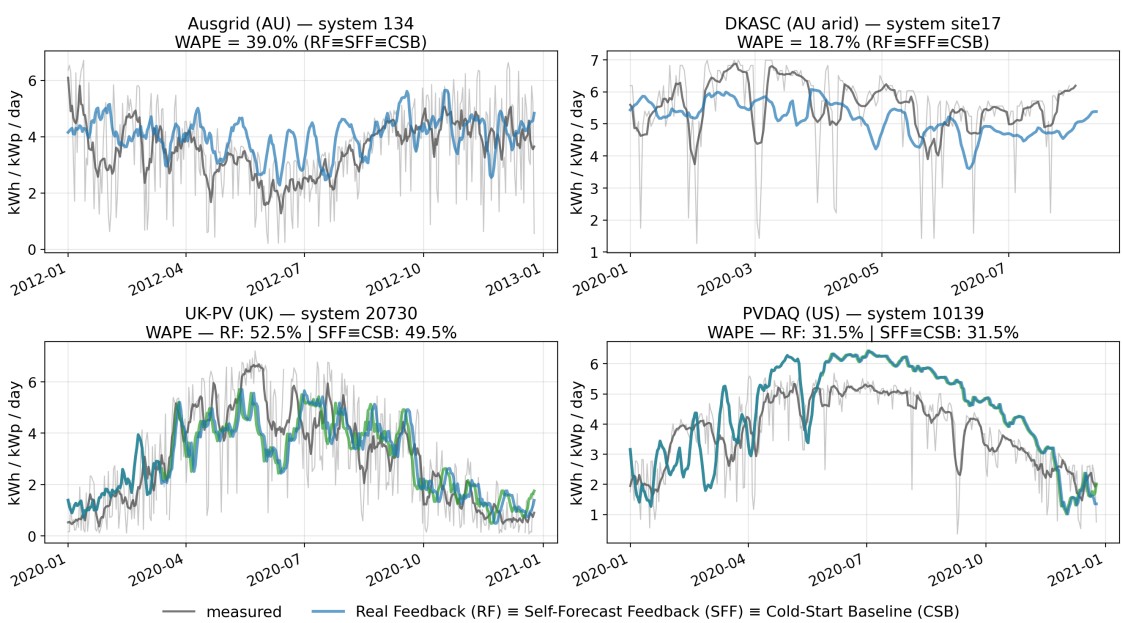

*Figure 7.* Seasonal-naive, PVGIS synthetic context.

### D.3. Prophet

Decomposable additive model (trend, yearly seasonality, regression on past meteorology), fit once per system on synthetic training data and not re-fit at rolling origins (consistent with the zero-shot protocol). At inference it uses only the calendar date and horizon meteorological covariates, so the strategy-specific context tail (real measurements in RF, self-forecasts in SFF, none in CSB) is never consumed. Consequently, the three traces overlap by construction; the residual $0.1 \, \mathrm{pp}$ WAPE gap arises from evaluation coverage (CSB spans the full year, whereas the rolling protocol drops the trailing partial window). This invariance acts as an anti-leakage check for the rolling-origin protocol. The absolute level is determined entirely by the synthetic training history, explaining the offset between OPAQUE and PVGIS panels and the residual bias relative to measurements where site-specific shading or soiling are not captured.

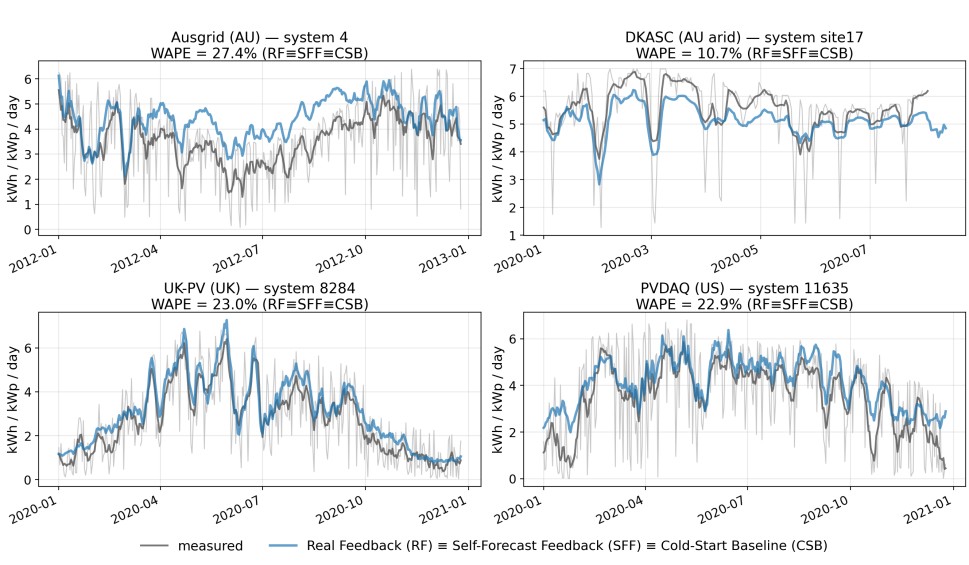

*Figure 8.* Prophet, OPAQUE synthetic context. RF and SFF coincide bit-for-bit; CSB coincides on the days it shares with the rolling protocol (cf. subsection text).

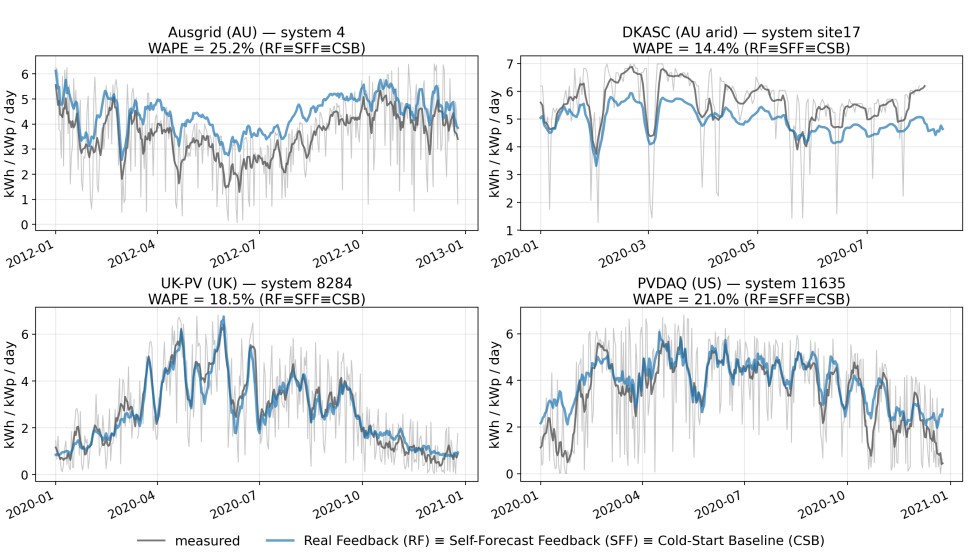

*Figure 9.* Prophet, PVGIS synthetic context.

## D.4. Chronos-2

T5 encoder-decoder foundation model. The encoder reads target-side context, the decoder generates the horizon attending to encoder output and future meteorological covariates. Under RF, the trace tracks measured seasonality closely on every cohort; under SFF and CSB, the decoder reverts toward the climatological mean of the synthetic source with positive bias on Ausgrid and PVDAQ, where the simulator does not capture site-specific shading, soiling, and degradation effects.

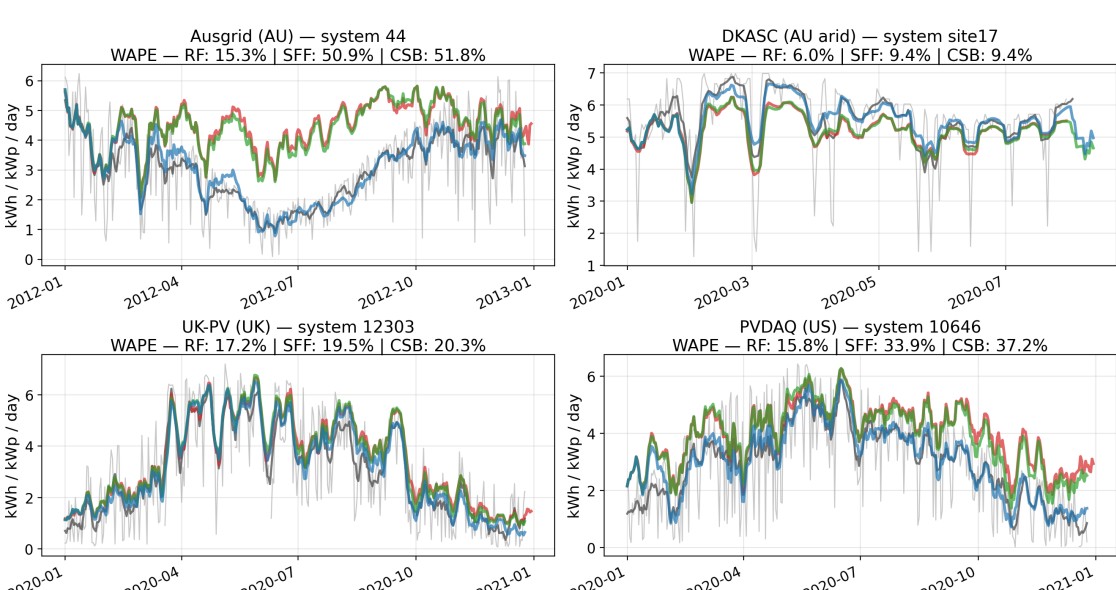

*Figure 10.* Chronos-2, OPAQUE synthetic context.

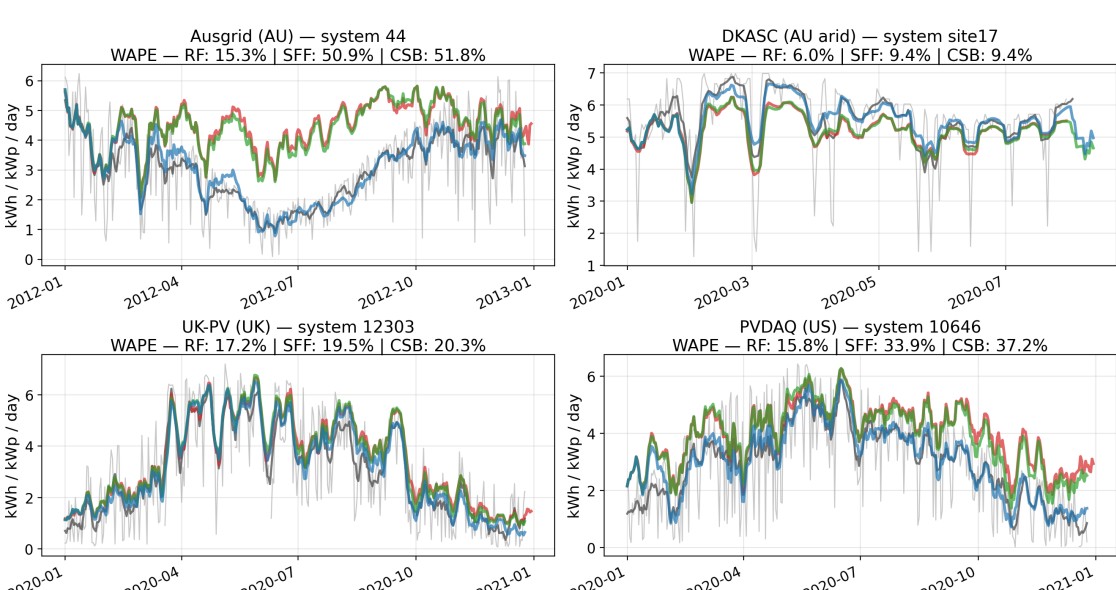

*Figure 11.* Chronos-2, PVGIS synthetic context.

## D.5. Moirai 2.0

Decoder-only Transformer with single-patch tokenisation, quantile loss, and multi-token prediction. Under RF the trace tracks the measured envelope on every cohort; under SFF and CSB the two traces converge to a nearly identical seasonal curve, indicating that the autoregressive rollout reaches a stable fixed-point that preserves the annual irradiance pattern rather than collapsing to a flat climatological mean. The decoder-only design accounts for the markedly reduced $\Delta_{\text{SFF}}$ and $\Delta_{\text{CSB}}$ penalties (Table 7).

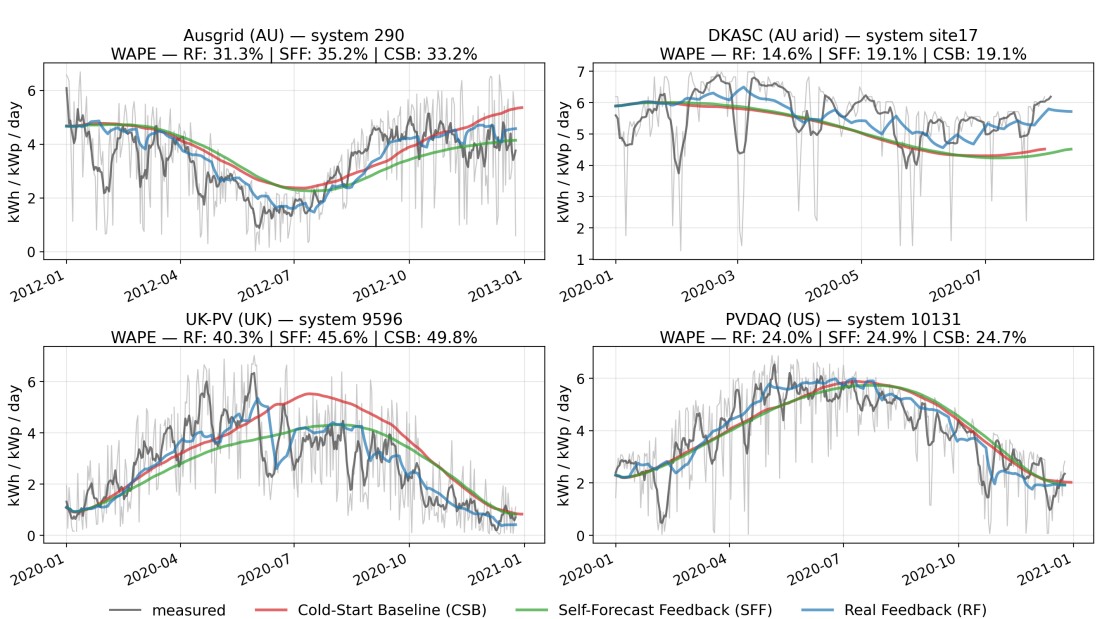

*Figure 12.* Moirai 2.0, OPAQUE synthetic context (SFF and CSB nearly indistinguishable).

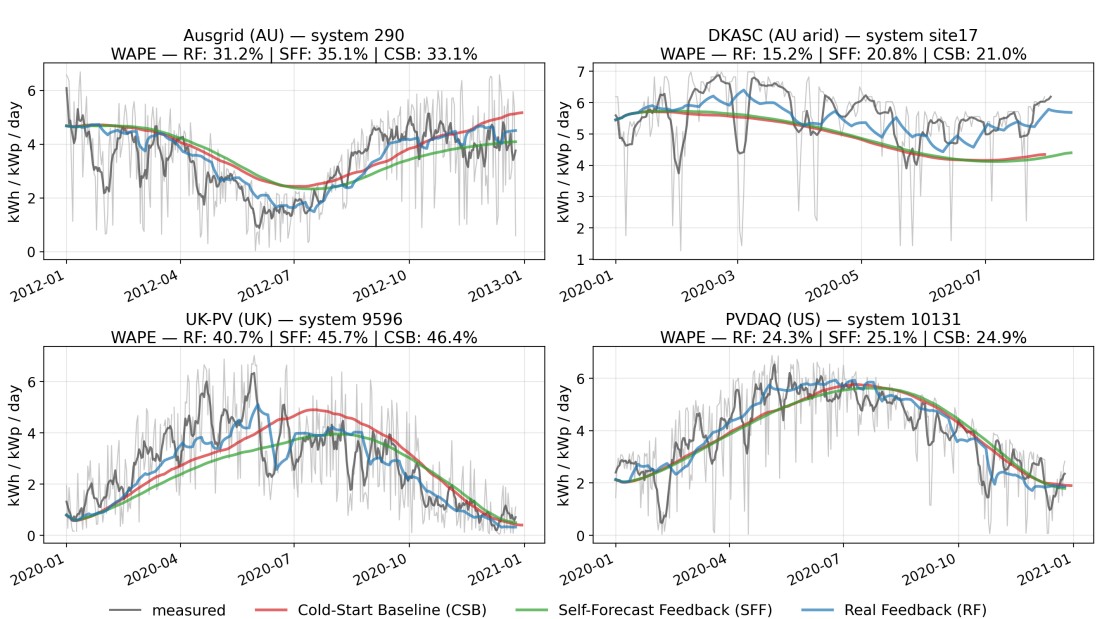

*Figure 13.* Moirai 2.0, PVGIS synthetic context.

## D.6. TimesFM 2.5

Decoder-only Transformer with patched inputs and a side-information channel for horizon covariates. The causal decoder leans on horizon covariates more than on target context, so under SFF and CSB the trace retains the seasonal envelope on the regular-irradiance cohorts (DKASC, UK-PV) and tracks the synthetic-source seasonality where the surrogate dominates the signal.

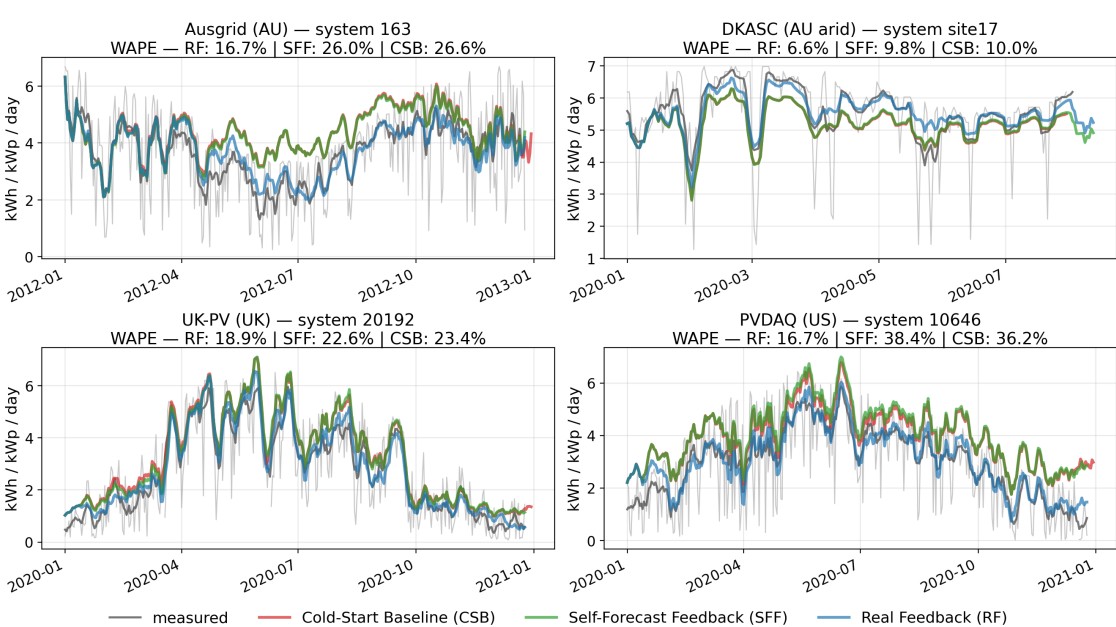

*Figure 14.* TimesFM 2.5, OPAQUE synthetic context.

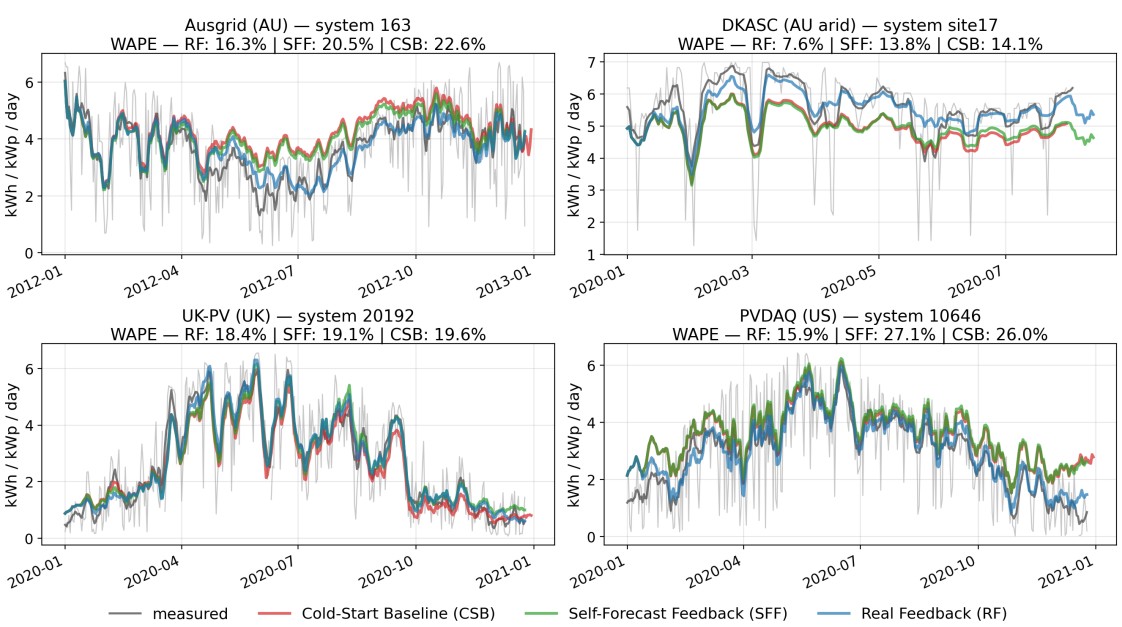

*Figure 15.* TimesFM 2.5, PVGIS synthetic context.

## D.7. TiRex

xLSTM foundation model, the only univariate forecaster of the five TSFMs: by API design it ignores meteorological covariates, both past and at the horizon. Under RF the autoregressive signal from recent measurements suffices; under SFF and CSB it has no covariate fallback and the trace collapses to the climatological mean (largest autoregressive penalty among foundation models in Table 7).

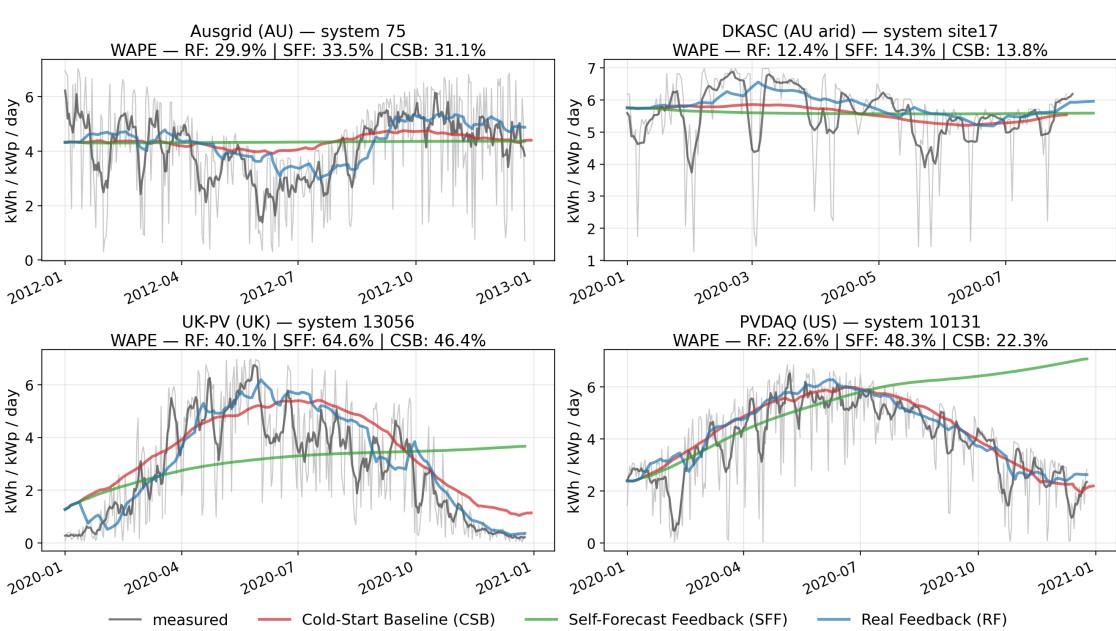

*Figure 16.* TiRex, OPAQUE synthetic context.

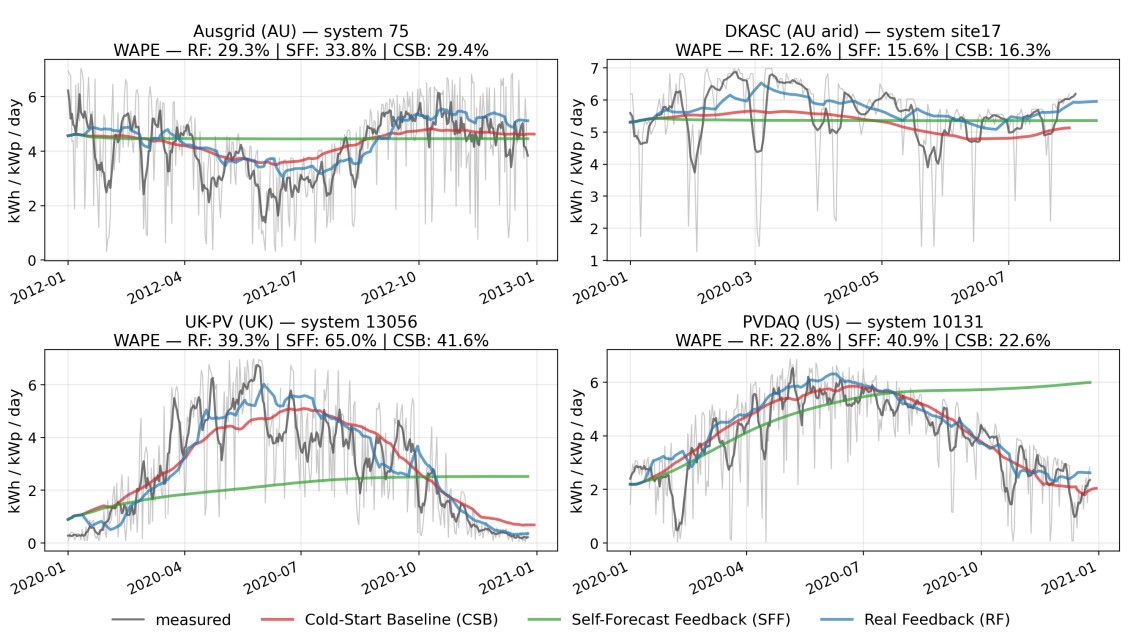

*Figure 17.* TiRex, PVGIS synthetic context.

## D.8. TabPFN-TS

Tabular in-context learner: prior-fitted network treating forecasting as tabular regression with engineered features (target-context lags, calendar variables, exogenous covariates), plus a native static-covariate interface ingesting plant metadata $\mathcal{M}$ (peak power, tilt, azimuth, technology). The static channel acts as a strategy-invariant anchor: under CSB the trace stays closest to the measured envelope of all foundation models. The same mechanism, however, makes TabPFN-TS the most sensitive backbone under SFF, since model self-forecasts enter the tabular feature vector and corrupt the regression input.

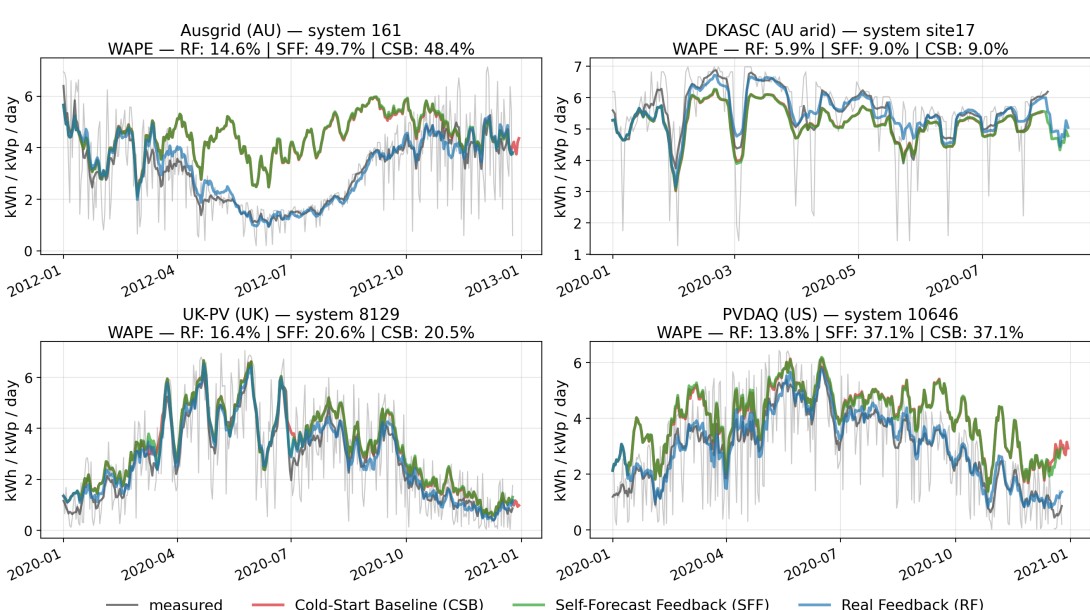

*Figure 18.* TabPFN-TS, OPAQUE synthetic context (CSB stays closest to measured).

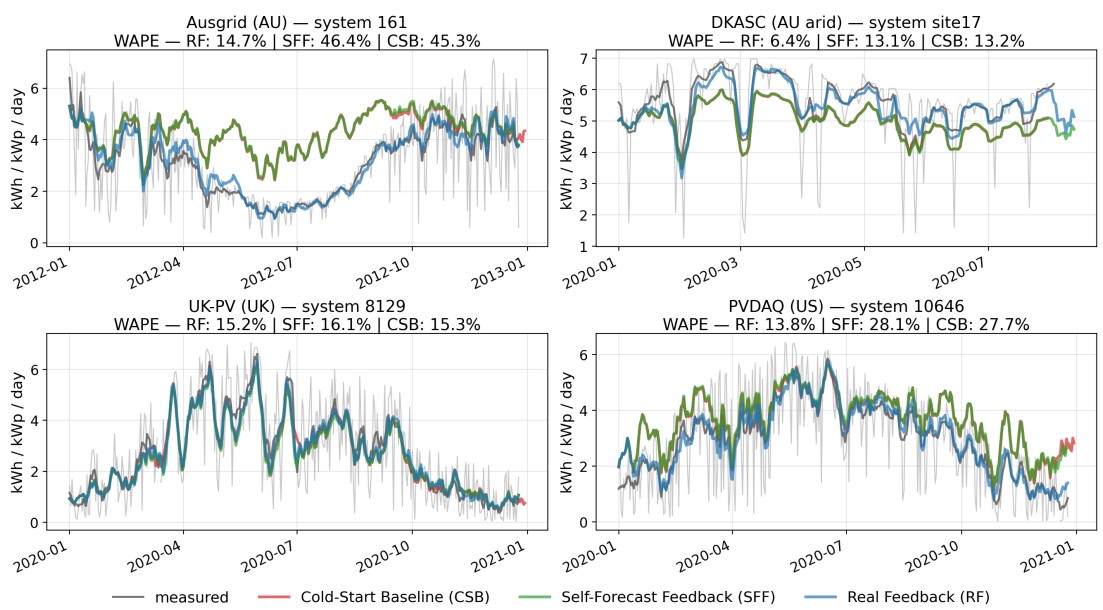

*Figure 19.* TabPFN-TS, PVGIS synthetic context.

