# OpenReview forum: "Time series Foundation Models based on Physics-Informed Synthetic Histories for Cold-Start Photovoltaic Forecasting"
_ICML.cc/2026/Workshop/FMSD — FMSD @ ICML 2026 Poster_

### Official Review · Reviewer_LDHS · 2026-05-19
**interesting paper but maybe with a different view**

**Rating:** 6
**Confidence:** 4

**Review:**

## Summary

This paper addresses the cold-start problem that arises when there are few or no data points available, in this case, for PV systems, but predictions for them are still required. To address this issue, the study examines the extent to which time series foundation models can help producing these forecast based on an synthetic context of the target time series together with weather covariates. Three different settings across a dataset containing worldwide PV systems  are investigated, yielding that models which can incorporate covariates like TabPFN or Chronos-2 perform best.

---

## Strengths
- making predictions with little or no data is a common scenario in practice, whether, as in this case, it involves new solar power plants, or any other newly initiated time series data
- clear set-up, well explained
- introducing a new synthetic data generator is ambitious for a workshop paper
- the core concept and method are well described
- suitable baselines with Prophet and seasonal-naive
- different settings under which the models are evaluated

---

## Areas for Improvement

- although I like that it is explained why future weather data is treated as future known covariate and it is common practice to do so, we still have in practice two distinct data types: historical weather data and forecast weather data for future data points. For real-world usage like this cold start case is framed, the weather forecast error is real and could affect the models differently and therefore yield different results, especially there is no serious weather forecast for a 365 day horizon. Unfortunately, this calls into question the relevance of the CSB setting, since the cold-start scenario is precisely a practical concern when a new system is being connected. And in practice, we don’t have weather data for a year in the future. For a future extended version of the paper I would recommend to focus either on shorter horizons where a weather forecast is reliable enough or to frame it differently. Additionally an ablation study on the models sensitivity on weather forecast errors would be beneficial.
- if I haven't overread it, it is only mentioned in the very last sentence on which granularity/frequency the evaluation is based. A finer resolutions like 15 min data together with day-ahead forecasts are highly relevant in the industry and would be an interesting addition.
- an analysis is missing if the curated data is already included in the pre-training corpus of the Time Series Foundation Models. Especially the Ausgrid data could be part of the Monash collection which is widely used in the community (unfortunately the website is currently down)
- another key factor: when generating synthetic context, why not using the same generator as baseline or as forecast? Forecasts based on purely synthetic context data shouldn't be able to outperform the generator, which is based on the same covariates the TSFM is seeing (otherwise it would be a very surprising result). In fact this rises the question why to use TSFM anyhow on cold start settings. In my opinion the most interesting setting is the RF setting and there the question: how fast are TSFMs adopting to the real values and outperforming their synthetic context generator and is synthetic context helping with little data or not. Currently we see only the other way round, if real data is helping.

---

## Detailed Comments


If the above weaknesses will be addressed in further versions and the observations are confirmed, I see the usage of synthetic context data as proxy for missing real data a good way mitigating data scarcity and other interesting applications of TSFMs.

---

## Justification of Score

The problem setting testing TSFMs on no or little data is highly relevant for their practical use cases. Although I see exactly in the problem setting several problems, there is another view on the paper which makes it highly interesting and relevant: the overall investigation of TSFMs leveraging covariates for way better results and as well the idea of mixture of synthetic and real data as context, make the paper a valuable basis for further discussions.

---

### Official Review · Reviewer_Ttcu · 2026-05-20
**Benchmark of TSFMs for cold-start PV forecasting with a useful open physics-based context generator**

**Rating:** 7
**Confidence:** 4

**Review:**

## Summary

This paper addresses cold-start photovoltaic (PV) forecasting, the setting where no historical production data exists at commissioning time. The authors propose a two-stage zero-shot pipeline: a physics-based synthetic history generator (OPAQUE, introduced here, or PVGIS as an alternative) provides temporal context, which is then consumed by a frozen TSFM at inference time. Five TSFMs and classical baselines (persistence, seasonal-naive, Prophet) are evaluated across 440 PV sites from four datasets spanning three continents under three context strategies: strict cold-start (CSB), real feedback (RF), and self-forecast feedback (SFF). The main findings are that covariate-aware TSFMs outperform baselines by ~1.7-2x under RF, and that performance is largely insensitive to the synthetic history source, suggesting plausible temporal context matters more than generator fidelity.

## Strengths

- Well-motivated and practically relevant problem. Cold-start PV forecasting is a real operational constraint, and the zero-shot pipeline framing is a clean and appropriate solution
- Comprehensive evaluation. 440 sites, four datasets, five TSFMs, two synthetic sources, three context strategies, and three metrics make for an impressive experimental scope
- The OPAQUE generator is a concrete contribution: an open, satellite-free, physics-based synthetic history generator that is immediately useful to practitioners and reproducible without proprietary data sources
- The synthetic-source insensitivity finding is actionable and non-trivial. The result that OPAQUE and PVGIS yield near-identical downstream performance is an important practical insight: any plausible physics-based simulator suffices, which lowers the barrier to adoption
- The three-strategy design (CSB / RF / SFF) is well thought out, cleanly separating the strict cold-start regime from post-commissioning telemetry availability and exposing complementary failure modes
- Good fit for the workshop, bridging TSFMs, domain-specific applications, and structured data with real-world constraints

## Areas for Improvement

- The use of perfect-foresight ERA5 inputs is a significant confound, acknowledged in the limitations but not quantified. How much of the TSFM advantage over Prophet collapses under realistic weather forecast inputs? This is the most important gap for operational validity
- Single-seed evaluation is flagged as a limitation; for models with stochastic inference (Chronos-2, Moirai 2.0) some measure of variance across seeds would strengthen confidence in the rankings

## Detailed Comments

- The claim of "1.7-2x lower error than Prophet" applies under RF, not CSB. The abstract and introduction could be clearer that this is a post-commissioning result rather than a strict day-0 result
- TabPFN-TS is the only model ingesting static plant metadata, which is an important architectural asymmetry that likely drives part of its RF advantage and should be noted more prominently in the main text discussion, not only in the appendix
- The covariate-consumption taxonomy (M1/M2/M3) in Appendix B is a useful framing that could be surfaced briefly in the main text to help readers interpret the model ranking
- Future work on synthetic-data-driven fine-tuning is well motivated given these results

## Justification of Score

This is a solid, well-executed application paper with a genuine domain contribution (OPAQUE), a thoughtful experimental design, and a clean actionable finding about synthetic context insensitivity. The main weaknesses are the perfect-foresight weather inputs and the absence of a fine-tuning comparison, but these are not disqualifying. Clearly above the acceptance threshold.

---

### Official Review · Reviewer_qAHD · 2026-05-21
**Time-Series Foundation Models based on Physics-Informed Synthetic Histories for Cold-Start Photovoltaic Forecasting**

**Rating:** 5
**Confidence:** 5

**Review:**

## Summary
This is an application of foundation models paper, which wants to make use of time-series foundation models for the cold-start photovoltaic forecasting problem. The authors propose a two-stage zero-shot pipeline: a synthetic-history generator (PVGIS or the new satellite-free OPAQUE chain built from pvlib with Hay-Davies, NOCT, and PVWatts) produces a year of plausible daily yield from commissioning-time metadata and ERA5 weather, and a downstream zero-shot TSFM (Chronos-2, Moirai 2.0, TimesFM 2.5, TiRex, TabPFN-TS) consumes that sequence as context and forecasts the held-out year. Five TSFMs and classical baselines (naive, seasonal-naive, Prophet) are benchmarked across 440 PV sites in 4 datasets (Ausgrid, DKASC, UK-PV, PVDAQ) under three context strategies: Cold-Start Baseline (CSB, synthetic only), Real Feedback (RF, replace synthetic with measured as it accumulates), and Self-Forecast Feedback (SFF). Headlines: covariate-aware FMs beat classical baselines roughly 1.7–2× under RF; TabPFN-TS leads under RF; Chronos-2 is most robust under SFF; the synthetic source (OPAQUE vs PVGIS) makes little difference on average.

## Strengths
1. The evaluation spans a variety of datasets (440 sites across Ausgrid, DKASC, UK-PV, and PVDAQ), which gives the rankings a reasonable amount of cross-cohort breadth.
2. The topic is relevant and fits the scope of the FMSD workshop: cold-start PV is a real deployment regime, and the synthetic-history-as-context framing is the natural one for zero-shot TSFMs that cannot be fine-tuned at day 0.
3. The writing of the paper and the structuring are good; the strategy decomposition (CSB / RF / SFF), the per-cohort breakdown, and the daily forecast-trace appendix are easy to follow.

## Weaknesses
1. The paper claims cold-start usage, but the data shows that on 0 days CSB (OPAQUE/PVGIS) the baseline Prophet is better than or close to the best foundation model on most cohorts; it is only after roughly 15 days of real data has accumulated under Real Feedback that the foundation models clearly beat the classical baselines. The claims could have been toned down accordingly to say that foundation models make best usage of a *less-data* situation rather than that they solve cold-start, since true day-0 performance is closer to parity than to dominance.
2. No multi-seed evaluation. Every reported number is a single-seed point estimate with no variance, which is insufficient for a benchmark whose central deliverables are model rankings (TabPFN-TS RF, Chronos-2 SFF, Moirai 2.0 SFF robustness); several close calls (TabPFN-TS RF MAE 0.514 vs Chronos-2 0.535) cannot be resolved from the data as written, and multi-seed evaluation with paired-Wilcoxon-on-sites tests is cheap given 440 sites.

## Questions
1. What is the value of T (forecast horizon, and the context length used at inference) for the reported numbers? The cold-start versus RF picture is sensitive to horizon, and the exact T should be pinned down at evaluation time.